# communications
# engineering

# Classifying seismograms using the FastMap algorithm and support-vector machines

Malcolm C. A. White [1✉], Kushal Sharma[2], Ang Li[2], T. K. Satish Kumar[2] & Nori Nakata[1,3]

Neural networks and related deep learning methods are currently at the leading edge of technologies used for classifying complex objects such as seismograms. However they generally demand large amounts of time and data for model training and their learned models can sometimes be difficult to interpret. FastMapSVM is an interpretable machine learning framework for classifying complex objects, combining the complementary strengths of FastMap with support vector machines (SVMs) and extending the applicability of SVMs to domains with complex objects. FastMap is an efficient linear-time algorithm that maps complex objects to points in a Euclidean space while preserving pairwise domain-specific distances between them. Here we invoke FastMapSVM as a lightweight alternative to neural networks for classifying seismograms. We demonstrate that FastMapSVM outperforms other state-of-the-art methods for classifying seismograms when train data or time is limited. We also show that FastMapSVM can provide an insightful visualization of seismogram clustering behaviour and thus earthquake classification boundaries. We expect FastMapSVM to be viable for classification tasks in many other real-world domains.

[1] Massachusetts Institute of Technology, Cambridge, MA 02139, USA. [2] University of Southern California, Los Angeles, CA 90007, USA. [3] Lawrence Berkeley National Laboratory, Berkeley, CA 94720, USA. ✉email: malcolmw@mit.edu

Various Machine Learning (ML) and Deep Learning (DL) methods, such as Neural Networks (NNs), are popularly used for classifying complex objects. For example, a Convolutional NN (CNN) is used for classifying Sunyaev-Zel'-dovich galaxy clusters[1], a densely connected CNN is used for classifying images[2], and a deep NN is used for differentiating the chest X-rays of Covid-19 patients from other cases[3]. However, they generally demand large amounts of time and data for model training; and their learned models can sometimes be difficult to interpret.

In this paper, we advance FastMapSVM—an interpretable ML framework for classifying complex objects—as a lightweight alternative to NNs for classification tasks in which train data or time is limited and a suitable distance function can be defined. While most ML algorithms learn to identify characteristic features of individual objects in a class, FastMapSVM leverages a domain-specific distance function on pairs of objects. It does this by combining the strengths of FastMap and support vector machines (SVMs). In its first stage, FastMapSVM invokes FastMap, an efficient linear-time algorithm that maps complex objects to points in a Euclidean space, while preserving pairwise domain-specific distances between them. In its second stage, it invokes SVMs and kernel methods for learning to classify the points in this Euclidean space.

The FastMapSVM framework that we implement in this paper is conceptually identical to the SupFM-SVM method[4], the application of which to complex objects was anticipated by the original authors. We present, to the best of our knowledge, the first such application to complex objects by using FastMapSVM to classify seismograms. We compare model performance against state-of-the-art NN alternatives in the seismogram domain using a benchmark data set to demonstrate the performance characteristics of FastMapSVM. We further demonstrate that FastMapSVM can be easily deployed for different real-world classification tasks in the seismogram domain. Our results motivate FastMapSVM as a potentially advantageous alternative to NNs in other domains when train data or time is limited and a suitable distance function can be defined. Additional results from a multi-class classification problem in a different domain (images of hand-written digits) are included as Supplementary Material (Suppl. Text S1 and Suppl. Figs. S3 and S4).

In this paper, we illustrate the advantages of FastMapSVM in the context of classifying seismograms. This is a particularly illustrative domain because seismograms are complex objects with subtle features indicating diverse energy sources such as earthquakes, ocean-Earth interactions, atmospheric phenomena, and human-related activities. We address two fundamental, perennial questions in seismology: (a) Does a given seismogram record an earthquake? and (b) Which type of wave motion, e.g., compressional (P-wave) versus shear (S-wave), is predominant in an earthquake seismogram? In Earthquake Science, answering these questions is referred to as "detecting earthquakes" and "identifying phases", respectively. The development of efficient, reliable, and automated solution procedures that can be easily adapted to new environments is essential for modern research and engineering applications in this field, such as in building Earthquake Early Warning Systems. Moreover, a model imposing modest demands on train data will aid the analysis of signal classes for which large train data sets are unavailable, such as "icequakes," stick-slip events at the base of landslides, and nuisance signals recorded during temporary seismometer deployments. Towards this end, we show that FastMapSVM is a viable ML framework. Through experiments, we show that FastMapSVM (a) outperforms state-of-the-art NNs for classifying seismograms when train data or time is limited, (b) can be rapidly deployed for different real-world classification tasks, and (c) is robust against noisy perturbations to model inputs.

The key contributions of this paper are as follows:

1. We present the first application of FastMapSVM to classifying complex objects—namely, seismograms—for which it is hard to extract features of individual objects but is easy to define a distance function on pairs of objects.
2. We demonstrate that FastMapSVM outperforms state-of-the-art NNs for classifying seismograms when train data or time is limited.
3. We discuss the sensitivity of FastMapSVM's performance to training parameters such as the size of the train data and the dimensionality of the Euclidean embedding.
4. We extrapolate our results to motivate FastMapSVM as a potentially advantageous, lightweight alternative to NNs for classifying complex objects in other domains as well when train data or time is limited.
5. We provide an efficient GPU-accelerated implementation of FastMapSVM, publicly available at: https://github.com/malcolmw/FastMapSVM.

## Results

**Data**. We assess the performance of FastMapSVM using seismograms from two data sets. All seismograms used in this paper record ground velocity at a sampling rate of 100 s$^{-1}$ and are bandpass filtered between 1 Hz and 20 Hz before analysis using a zero-phase Butterworth filter with four poles; we refer to this frequency band as our passband.

*Stanford Earthquake Data Set (STEAD)*. The first data set is the Stanford Earthquake Data Set (STEAD)[5], a benchmark data set with $>1.2 \times 10^6$ carefully curated, three-component (3C), 60 s seismograms for training and testing algorithms in Earthquake Science. We select a balanced subset of 65,536 3C seismograms from the STEAD comprising 32,768 "earthquake" and 32,768 "noise" seismograms. Earthquake seismograms record ground motions induced by a nearby earthquake; whereas noise seismograms record no known earthquake-related ground motions. We randomly select earthquake seismograms using a selection probability that is inversely proportional to the kernel density estimate of the 5-D joint distribution over (a) epicentral distance, (b) event magnitude, (c) event depth, (d) time interval between P- and S-wave arrivals, and (e) signal-to-noise ratio (SNR) (Suppl. Fig. S1). This scheme is designed to yield a broad distribution of seismograms. All earthquake seismograms are recorded by a seismometer within 100 km of the epicenter, have a hypocentral depth of less than 30 km, and have P- and S-wave arrival times manually identified by a trained analyst (no automated arrival picks). Distributions in this subset that are skewed towards shallow earthquakes with low magnitude and low SNR reflect the distribution of natural earthquakes and their recordings. Noise seismograms are randomly selected to maximize the diversity of geographic recording locations.

From this base data set of 65,536 seismograms, we draw a simple random sample (SRS) of 16,384 earthquake seismograms and an equal-sized SRS of noise seismograms for model training (the "train" data set). The 32,768 remaining seismograms make up the "test" data set. The test data are trimmed to 30 s per seismogram, including an amount of time uniformly distributed between 4 s and 15 s preceding the P-wave arrival for earthquake seismograms. Note that both train and test data sets are balanced across the earthquake and noise classes. We iteratively draw SRSs from the train data set to create multiple smaller, balanced train data sets, each of which is half the size of

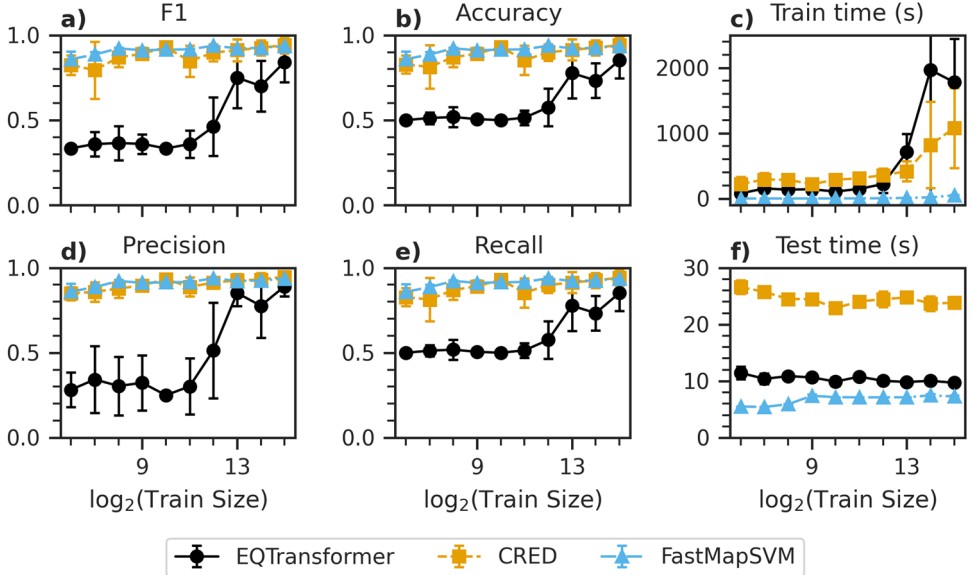

**Fig. 1 Model performance metrics with varying train data size.** Shows the performance of EQTransformer (black circles and solid line), CRED (orange squares and dashed line), and FastMapSVM (light blue triangles and dash-dotted line) on the STEAD with varying train data size. **a–f** The F1 score, accuracy, train time, precision, recall, and test time, respectively. Error bars represent the standard deviation of the measurements over 20 trials for FastMapSVM and 10 trials for EQTransformer and CRED. The recall is identical to the accuracy in our case.

the sample from which it was drawn. Thus, we have nested, balanced train data sets with sample sizes $2^n$ for integer $n$ between 6 and 15.

To assess model performance for identifying phases, we select an independent subset of 538 3C, 3 s seismograms from the STEAD, all of which are recorded by station TA.109C; 269 seismograms start 1 s before a P-wave phase arrival, and 269 seismograms start 1 s before an S-wave phase arrival.

*Ridgecrest data set.* The second data set, which we simply refer to as the "Ridgecrest" data set, comprises data recorded by station CI.CLC of the Southern California Seismic Network (SCSN) on 5 July 2019, the first day of the aftershock sequence following the 2019 Ridgecrest, CA, earthquake pair and on 5 December 2019, five months after the mainshocks. We use the earthquake catalog published by the Southern California Earthquake Data Center (SCEDC) to extract 512 3C, 8 s seismograms, 256 of which record both P- and S-wave phase arrivals from a nearby aftershock (between 4.5 km and 27.6 km epicentral distance), and the remaining 256 of which record only noise. All 512 of these signals are recorded on 5 July 2019. Earthquake magnitudes represented in the Ridgecrest data set range between 0.5 and 4.0, earthquake depths range between 900 m above sea level and 9.75 km below sea level, and SNRs range between −8 dB and 73 dB. The maximum peak ground acceleration recorded in the Ridgecrest data set is 0.197 m/s².

We use the Ridgecrest data set to first demonstrate the robustness of FastMapSVM against noisy perturbations. We then use it to demonstrate FastMapSVM's ability to detect new microearthquakes by automatically scanning a 24 h, continuous, 3C seismogram recorded between 00:00:00 and 23:59:59 (UTC) on 5 December 2019. Whereas the analysis on the STEAD demonstrates FastMapSVM's performance on a benchmark, the analysis on the Ridgecrest data set provides an example of a more realistic use case of FastMapSVM: After handpicking a small number of earthquake and noise signals—a task that even a novice analyst can perform in a few hours—continually arriving seismic data can be automatically scanned for additional earthquake signals. This capability manifests the primary conclusion of

the preceding robustness test: Even when earthquake signals are difficult to discern by the human eye, FastMapSVM can often reliably detect them.

### STEAD analysis

*Detecting earthquakes in the STEAD.* The *EQTransformer* DL model[6] for simultaneously detecting earthquakes and identifying phase arrivals is arguably the most accurate, publicly available model for this pair of tasks. The authors of EQTransformer report perfect precision and recall scores for detecting earthquakes in 10% of the STEAD waveforms after training its roughly $3.72 \times 10^5$ model parameters with 85% of the STEAD waveforms; 5% of the STEAD waveforms were reserved for model validation. (Note that the authors of EQTransformer used a version of the STEAD with $1 \times 10^6$ and $3 \times 10^5$ earthquake and noise waveforms, respectively, which differs slightly from the newer version of the STEAD we use.) The *CRED* model[7] is another DL model for detecting earthquakes, which scored perfect precision and 0.96 recall using the same train and test data as EQTransformer[6]. The CRED model does not identify phase arrivals. We choose these two DL models for comparison because EQTransformer is popularly used[8,9] and represents the state-of-the-art in general practice, CRED is designed for the sole task of detecting earthquakes (i.e., does not simultaneously identify phases), and the pre-trained models are readily available through the SeisBench[10] package.

To compare the performance of FastMapSVM against EQTransformer and CRED in the case of limited train data, we train multiple instances of each model using different amounts of train data and test them on the same test data (the 32,768 test seismograms selected from the STEAD, as described above). All models are trained and tested using an NVIDIA RTX A6000 GPU. We train and test each model multiple times for each train data size to estimate statistics for performance scores (F1, accuracy, precision, and recall; Fig. 1a, b, d, e), train time (Fig. 1c), and test time (Fig. 1f). Performance scores are averaged over both labels. We repeat 20 trials for each train data size for FastMapSVM but limit the number of trials for EQTransformer and CRED to 10 because training becomes prohibitively time-

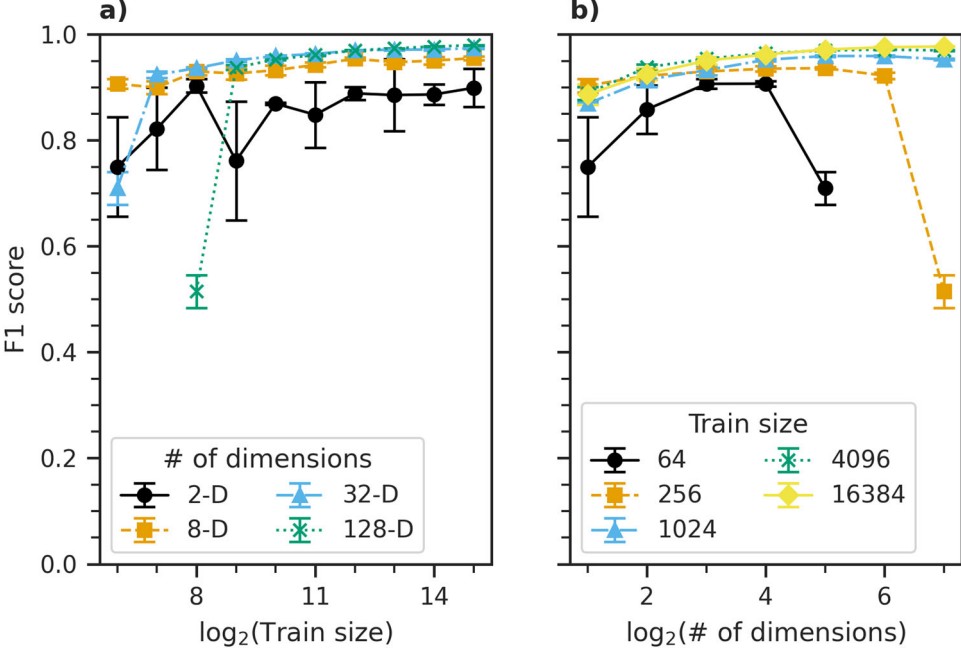

**Fig. 2 FastMapSVM's sensitivity to train data size and dimensionality of Euclidean embedding.** Shows the F1 score for varying train data size and dimensionality of Euclidean embedding. **a, b** These results for train data size and dimensionality of Euclidean embedding on the horizontal axis, respectively. Error bars represent the standard deviation of the measurements over 20 trials. Marker and line colors in (**a**) represent the number of dimensions of the embedding. Marker and line colors in (**b**) represent the number of instances in the train data set.

consuming for large train data sizes. The FastMapSVM model used here comprises a four-dimensional Euclidean embedding. Train data for FastMapSVM are trimmed to 30 s per seismogram, including 4 s of data preceding the P-wave arrival for earthquake seismograms. Unless otherwise noted, we set the probability threshold for the decision boundary to 0.5 in this paper.

FastMapSVM consistently outperforms EQTransformer and CRED using less train time for all train data sizes. FastMapSVM train times are 1–3 orders of magnitude smaller than those for EQTransformer and CRED. The respective performances of EQTransformer and CRED approach that of FastMapSVM as the train data size increases; however, they do so at the cost of rapidly increasing train times. The prediction times for EQTransformer and CRED are 1.33–2.08 and 3.17–4.82 times the prediction time for FastMapSVM, respectively. FastMapSVM also exhibits more stable performance than the DL models (i.e., less variance between trials), because the final performance of the DL models is sensitive to the random initial values of the model parameters.

The performance of FastMapSVM can be further improved by increasing the dimensionality of the Euclidean embedding, as demonstrated below; however, the prediction and train time both increase with the dimensionality of the embedding. This increase owes primarily to the number of calls to the user-supplied distance function, which increases linearly with the dimensionality of the Euclidean embedding. Thus, efficient distance functions need to be designed to maintain end-to-end computational efficiency for models with high-dimensional Euclidean embeddings.

*Sensitivity to train data size and dimensionality of Euclidean embedding.* Two important, interdependent questions concerning FastMapSVM follow: (a) How much train data is needed to train the model? and (b) How many Euclidean dimensions are needed to represent the objects being classified? To address these questions, we obtain a suite of models for different train data sizes and dimensionalities of Euclidean embedding. For each pair of train

data size and dimensionality of Euclidean embedding, we repeat 20 trials of model training and testing to obtain estimates of F1 score statistics (Fig. 2)—we choose the F1 score here because it is sensitive to both model precision and recall. Furthermore, recall is identical to accuracy in this analysis. All models are tested on the same test data used in the preceding experiments.

Model performance generally increases with, and exhibits the property of diminishing returns with respect to, train data size (Fig. 2a). This property is favorable because strong model performance can be achieved with very small train data sizes. For example, models trained on only 64 seismograms—a small fraction of what typical DL models require—achieve average F1 scores as high as 0.91. All models perform poorly when the number of train instances per class is equal to the number of Euclidean dimensions, and large models perform poorer than small models in these cases. A significant improvement is seen, however, when the train data size per class is at least twice the number of dimensions. Thus, models should be trained using a number of instances per class that is at least twice the number of dimensions.

Model performance also generally increases with, and exhibits the property of diminishing returns with respect to, dimensionality of Euclidean embedding (Fig. 2b). This property is favorable because small models yield rapid predictions and can be trained quickly using small data sets. For example, an eight-dimensional model trained with only 512 train instances achieves an average F1 score of 0.93. Furthermore, this property is attractive from the perspectives of memory consumption and data visualization in low-dimensional Euclidean space.

Models comprising only two Euclidean dimensions achieve average F1 scores as high as 0.90; however, the performance of such small models varies significantly, and does not increase with train data size reliably. Models comprising four dimensions achieve average F1 scores as high as 0.93 and exhibit significantly less variability than those comprising only two. These results suggest that models should comprise at least four dimensions for

most real-world applications. Moreover, trade-offs between model train time, train data size, prediction time, and performance make models comprising larger embeddings optimal in different circumstances. Models comprising more than 32 dimensions yield marginal improvements in this application, but take significantly longer to make predictions. Thus, we recommend users start with a number of model dimensions between 4 and 32 when training FastMapSVM in other application domains.

*Identifying phase arrivals.* As another illustration designed to demonstrate the effectiveness of FastMapSVM, we use the subset of 538 three-second seismograms recorded by station TA.109C from the STEAD to train and test a suite of models for classifying P- and S-wave phase arrivals (Fig. 3). We conduct 100 trials in which, for each trial, we select a balanced SRS of 268 seismograms for model training and test the model on the remaining 270 seismograms. Average F1, accuracy, precision, and recall scores are all above 0.89. Although these scores are relatively modest in comparison to those of state-of-the-art NNs designed for similar tasks and trained using much larger data sets, they demonstrate that FastMapSVM can be easily trained for strong performance on different classification tasks using only small amounts of time and data.

### Ridgecrest analysis

*Robustness against noisy perturbations.* It is important that a classification framework is robust against noisy perturbations of inputs. In general, the robustness of FastMapSVM against noisy perturbations may depend on the characteristics of the data and the chosen distance function. For classifying seismograms, we demonstrate FastMapSVM's robustness against noisy perturbations made to the Ridgecrest data set using the distance function described in the Methods section. We conduct 100 trials in which, for each trial, we randomly select 8 earthquake signals and 8 noise signals to train a FastMapSVM model with a four-dimensional Euclidean embedding; the 496 remaining seismograms are used for model testing. Each test instance is circularly shifted by an offset (in seconds) chosen uniformly at random from the interval $[−2, 2]$. FastMapSVM classifies test seismograms with an average accuracy of $(0.995 \pm 0.001)$. We subsequently conduct a set of experiments in which each model's performance is scored after perturbing signals in the test data set with increasing amounts of Gaussian noise. First, all test signals are normalized by their

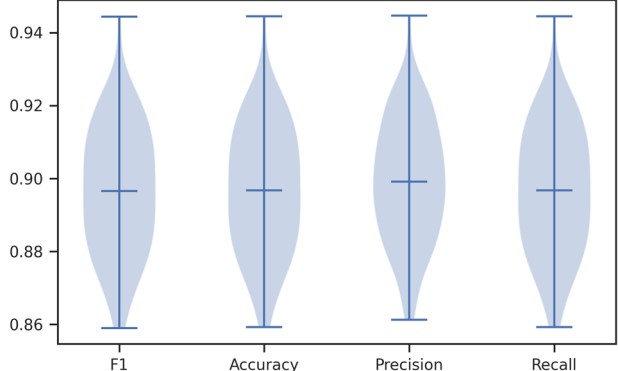

**Fig. 3 FastMapSVM's performance on identifying phases for station TA.109C in STEAD.** Shows the "violin plot" for F1, accuracy, precision, and recall score distributions obtained by the learned model for classifying P- and S-wave phase arrivals at station TA.109C in the STEAD over 100 trials. The bottom, middle, and top horizontal bars represent the minimum, mean, and maximum values, respectively, for each distribution. Shaded blue regions show kernel density estimates for the distributions.

standard deviation. Then, for each trial, we perturb each signal in the test data set by adding Gaussian noise with zero mean and standard deviation $\sigma$; $\sigma$ increases by 0.5 after each trial. Figure 4a shows how a waveform changes with increasing $\sigma$. Figure 4b shows the performance of FastMapSVM with increasing $\sigma$. FastMapSVM continues to classify seismograms with high accuracy and precision, even as earthquake signals become indiscernible to the human eye; e.g., the FastMapSVM model achieves 0.96 accuracy and precision scores for $\sigma = 2$. These results encourage us to deploy FastMapSVM in noisy environments to detect low-amplitude signals.

Counterintuitively, the model correctly labels earthquakes regardless of the amplitude of the noisy perturbations. Moreover, the model misclassifies noise signals as earthquakes (false positive errors) more frequently when the amplitude of the noisy perturbations is increased. With enough added noise, the model classifies all signals as earthquakes. Because we report performance scores averaged over both labels, this phenomenon is not obvious in Fig. 4.

The unique frequency content of the noisy perturbations (Suppl. Fig. S2) is responsible for the bias towards false positives in these experiments. In our passband, the average frequency spectrum of earthquake signals is nearly flat; whereas the average frequency spectrum of authentic seismic noise signals has prominent peaks near the low- and high-frequency endpoints. Because the noisy perturbations are Gaussian, their frequency spectrum is flat. This makes the spectra of noisy perturbations more similar to those of earthquake signals than to those of authentic seismic noise signals.

*Automatic scanning.* We further demonstrate a use case-inspired application of FastMapSVM using a 32-dimensional model trained on 256 earthquake signals and 256 noise signals selected randomly from the Ridgecrest data set. We apply the model to 8 s windows extracted from a 24 h continuous, 3C seismogram with 25% overlap and register detections for windows with detection probability > 0.95. The test seismogram was recorded by station CI.CLC between 00:00:00 and 23:59:59 (UTC) on 5 December 2019. We also apply the pre-trained EQTransformer[6] and CRED[7] models on the same data.

For each detection, we compute two quantities: (1) the maximum SNR and (2) the maximum normalized cross-correlation coefficient measured against the 256 earthquake seismograms used to train FastMapSVM (Fig. 5). We define the SNR as

$$\text{SNR} = 10 \log_{10}\left(\frac{P_{signal}}{P_{noise}}\right), \tag{1}$$

where $P_{signal}$ and $P_{noise}$ represent the average power of the signal and noise, respectively, which are measured in 1 s and 10 s sliding windows, respectively.

CRED registers the largest number of detections (1831), EQTransformer registers the fewest (805), and FastMapSVM registers an intermediate number (1453). Although CRED registers the largest number of detections, a large proportion of them correspond to very low SNR (<2.5) signals with low normalized cross-correlation coefficients (<0.2), which implies that a large proportion of them are likely false detections. Indeed, visual inspection of an SRS of these detections confirms this. FastMapSVM also registers a significant number of detections with relatively low SNR (<5); however, these detections are generally associated with higher normalized cross-correlation coefficients (>0.2) and slightly higher SNR (>2.5).

The majority of detections registered by FastMapSVM are associated with low to moderate SNR (between 2.5 and 10) and

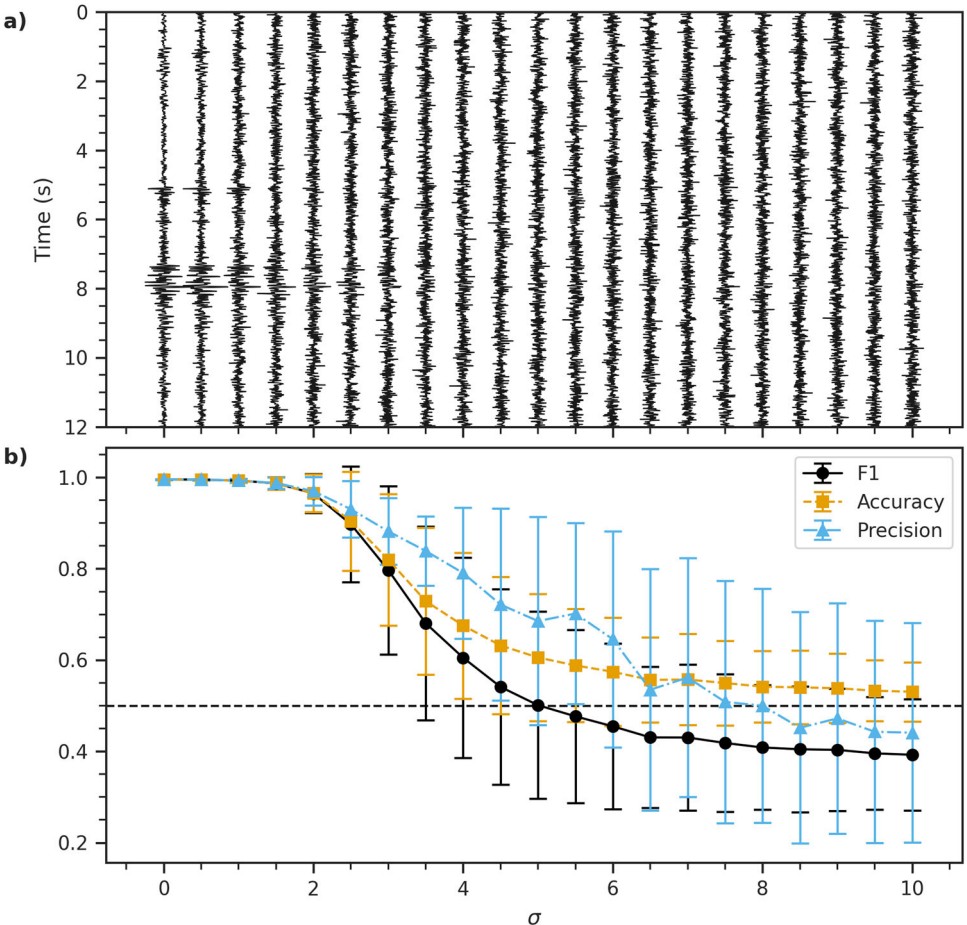

**Fig. 4 FastMapSVM's robustness against noisy perturbations.** Shows the performance of FastMapSVM on the Ridgecrest data set. **(a)** shows how a sample test waveform changes with the addition of increasing levels of Gaussian random noise with zero mean and standard deviation $\sigma$. It uses a vertical time-axis and an increasing $\sigma$ on the horizontal axis. Each waveform is self-normalized for plotting. **(b)** shows how the F1 (black circles and solid line), accuracy (orange squares and dashed line), and precision (light blue triangles and dash-dotted line) scores vary with increasing $\sigma$. Error bars represent the standard deviation of the measurements over 100 trials. The recall is omitted from this plot because recall and accuracy are identical in this case.

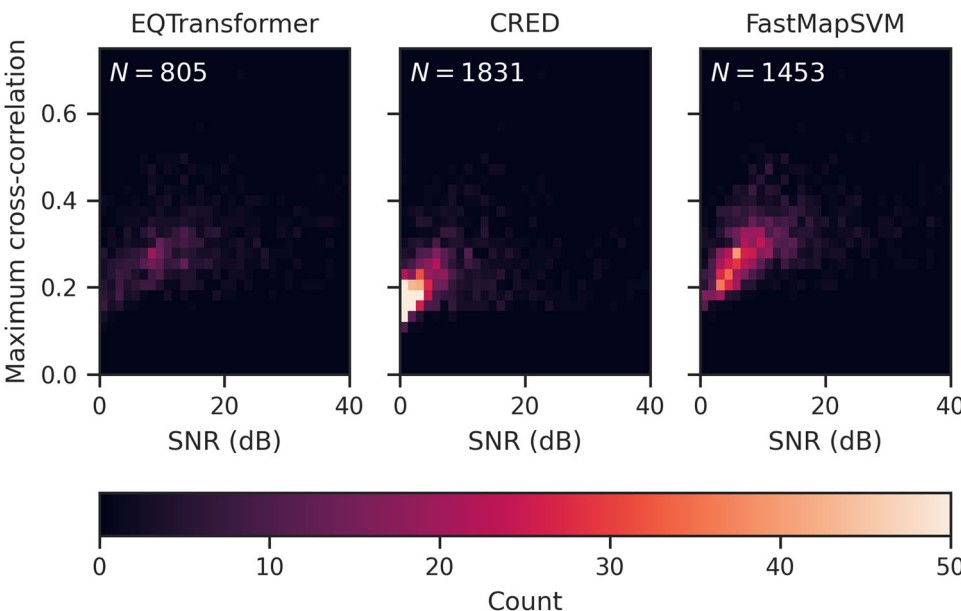

**Fig. 5 Comparison of automatic scanning results produced by EQTransformer, CRED, and FastMapSVM.** Shows the empirical joint distribution of maximum SNR (dB) and normalized cross-correlation coefficient for detections produced by **(left)** EQTransformer, **(middle)** CRED, and **(right)** FastMapSVM in an automatic scan of 24 h of data.

normalized cross-correlation coefficients (between 0.2 and 0.4). This is an expected consequence of the Gutenberg-Richter statistics that describe earthquake magnitude-frequency distributions. Perhaps surprisingly, FastMapSVM also registers a greater number of detections with high SNR (>10) than both EQTransformer and CRED. Visual inspection confirms that FastMapSVM seldom misses a high-SNR event detected by EQTransformer or CRED, whereas EQTransformer and CRED do occasionally miss high-SNR events detected by FastMapSVM.

Results presented in Fig. 5 suggest that (1) EQTransformer has the lowest detection and false detection rates, (2) CRED has the highest detection and false detection rates, (3) FastMapSVM has relatively high detection and low false detection rates, and (4) FastMapSVM detects high-SNR events with greater fidelity than EQTransformer and CRED.

## Discussion

Our use of FastMapSVM to classify seismograms is, to the best of our knowledge, the first application of the algorithm to complex objects for which a combination of representation and classification learning is necessary. Whereas prior work[4] focused on the sparse kernel representations enabled by FastMapSVM, our work focuses on the applicability of FastMapSVM in a real-world domain in which the objects are overwhelmingly complex for regular SVMs to be effective. In this case, we leverage a domain-specific distance function on pairs of seismograms, which is easier to build than identifying the subtle features of individual seismograms. Intuitively, FastMap is used as a "representational counterpart" to SVMs in FastMapSVM. Therefore, we focus on illustrating the benefits of FastMapSVM in a complex domain where a combination of representation and classification learning is required. Moreover, we compare FastMapSVM against other ML methods, namely, NNs, that seamlessly integrate the tasks of representation and classification learning. In this section, we discuss some of the advantages of FastMapSVM over existing ML methods for classifying complex objects such as seismograms. We base our discussion on our results in the specific context of classifying seismograms and generalize to the broader contexts of ML and data visualization.

Many existing ML algorithms for classification do not leverage domain knowledge when used off the shelf. Although a domain expert can occasionally incorporate domain-specific features of the objects being classified into the classification task, doing so becomes increasingly difficult as the complexity of the objects increases. FastMapSVM enables domain experts to incorporate their domain knowledge via a distance function instead of relying on complex ML models to infer the underlying structure in the data entirely. In fact, in many real-world domains, it is easier to construct a distance function on pairs of objects than it is to extract features of individual objects. Examples include DNA strings, images, and text documents, for which the edit distance, Minkowski distance[11], and cosine similarity[12], respectively, are well defined. Extracting features of individual objects is challenging in all of these domains, as in our seismogram domain. In the seismogram domain, our a priori knowledge that earthquake seismograms typically bear similarities to one another is encapsulated in a distance function that quantifies the normalized cross-correlation of the waveforms. This distance metric closely resembles other similarity metrics that have been extensively used in previous works in the Earthquake Science community[13–16]. FastMapSVM's strong performance owes partly to this incorporation of domain knowledge.

In addition, many existing ML algorithms produce results that are hard to interpret or explain. We define model *interpretability* as the degree to which causal mappings between model inputs

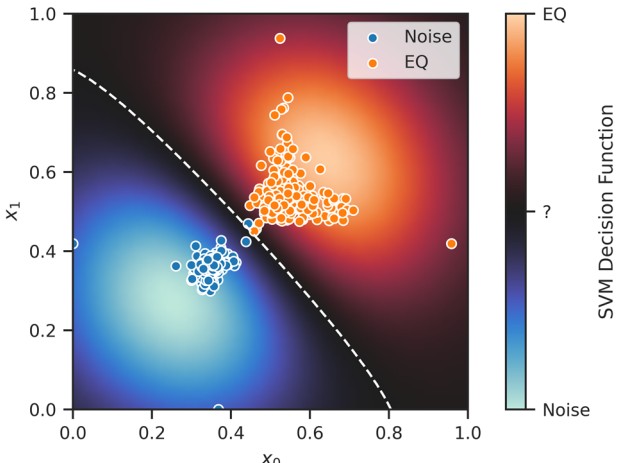

**Fig. 6 Perspicuous visualization of seismograms and classification boundaries produced by FastMapSVM.** Shows a visualization of FastMapSVM's classification boundary (dashed, white curve) and decision function (background) in a two-dimensional Euclidean embedding of the train data from the Ridgecrest data set. EQ refers to earthquakes. Warm and cold colors represent regions of high-confidence for earthquake and noise labels, respectively. Black background represents regions of relatively high uncertainty.

and outputs can be understood by humans[17]. For example, in NNs, a large number of interactions between neurons with nonlinear activation functions makes a meaningful interpretation of causal mappings challenging. FastMapSVM, on the other hand, embeds complex objects in a Euclidean space by considering only the distance function defined on pairs of objects, despite the complexity of the objects themselves. In effect, it simplifies the description of the objects by assigning Euclidean coordinates, i.e., points in Euclidean space, to them with a simple interpretation: Points that are close in the Euclidean sense represent objects that are similar in the sense defined by the domain-specific distance function. Moreover, because the distance function is itself user-supplied and encapsulates domain knowledge, it can often be intuitively understood, as is the case for our cross-correlation distance in the seismogram domain. Our discussion of the reasons for the disproportionate number of false positives produced by FastMapSVM in the Robustness against noisy perturbations section provides an example of model interpretability in action. Moreover, FastMapSVM provides a perspicuous visualization of the objects and the classification boundaries between them (Fig. 6). FastMapSVM produces such visualizations very efficiently because it invests only linear time in generating the Euclidean embedding.

FastMapSVM uses significantly smaller amounts of time and data for model training compared to the NN models we tested. We demonstrate this key feature of FastMapSVM in the seismogram domain and expect that these results will generalize to other domains. Whereas NNs and other ML algorithms store abstract representations of the train data in their model parameters, FastMapSVM stores explicit references to some of the original objects, referred to as pivots. When making predictions, test instances are compared directly to the pivots using the user-supplied distance function. FastMapSVM thereby obviates the need to learn a complex transformation of the input data and thus significantly reduces the amount of time and data required for model training. Moreover, given $N$ train instances, FastMapSVM leverages $O(N^2)$ pieces of information via the distance function, which is defined on every pair of objects. In contrast, ML algorithms that focus on individual objects leverage only $O(N)$ pieces

of information. The efficiency of FastMapSVM is demonstrated by our results from the seismogram domain.

In general, FastMapSVM extends the applicability of SVMs and kernel methods to domains with complex objects. With increasing complexity of the objects, deep NNs have gained more popularity compared to SVMs because it is unwieldy for SVMs to represent all the features of complex objects in Euclidean space. FastMapSVM, however, revitalizes the SVM approach by leveraging a distance function to create a low-dimensional Euclidean embedding of the objects.

Overall, any application domain hindered by a paucity of train data but possessing a well-defined distance function on pairs of its objects can benefit from the advantages of FastMapSVM. Examples of such applications in Earthquake Science include analyzing and learning from data obtained by distributed acoustic sensing technology or during temporary deployments of "large-N" nodal arrays. Furthermore, the efficiency of FastMapSVM makes it suitable for real-time deployment, which is critical for engineering Earthquake Early Warning Systems.

## Conclusions

In this paper, we advance FastMapSVM—an interpretable ML framework that combines the complementary strengths of FastMap and SVMs—as an advantageous, lightweight alternative to existing methods, such as NNs, for classifying complex objects when train data or time is limited. FastMapSVM offers several advantages. First, it enables domain experts to incorporate their domain knowledge using a distance function. This avoids relying on complex ML models to infer the underlying structure in the data entirely. Second, because the distance function encapsulates domain knowledge, FastMapSVM naturally facilitates interpretability and explainability. In fact, it even provides a perspicuous visualization of the objects and the classification boundaries between them. Third, FastMapSVM uses significantly smaller amounts of time and data for model training compared to other ML algorithms. Fourth, it extends the applicability of SVMs and kernel methods to domains with complex objects.

We demonstrated the efficiency and effectiveness of FastMapSVM in the context of classifying seismograms. On the Stanford Earthquake Data Set, we showed that FastMapSVM performs comparably to state-of-the-art NN models in terms of precision, recall, and accuracy. It also uses significantly smaller amounts of time and data for model training compared to other methods and can yield faster predictions. On the Ridgecrest data set, we first demonstrated the robustness of FastMapSVM against noisy perturbations. We then demonstrated its ability to reliably detect new microearthquakes that are otherwise difficult to detect.

In future work, we expect FastMapSVM to be viable for classification tasks in many other real-world domains. In Earthquake Science, we will apply FastMapSVM to analyze and learn from data obtained during temporary deployments of large-N nodal arrays and distributed acoustic sensing. In Computational Astrophysics, we anticipate the use of FastMapSVM for identifying galaxy clusters based on cosmological observations. In general, the efficiency and effectiveness of FastMapSVM also make it suitable for real-time deployment in dynamic environments in applications such as Earthquake Early Warning Systems.

Our implementation of FastMapSVM is publicly available at: https://github.com/malcolmw/FastMapSVM.

## Methods

Our FastMapSVM method comprises two main components: (1) The FastMap algorithm[18] for embedding complex objects in a Euclidean space using a distance function, and (2) SVMs for classifying objects in the resulting Euclidean space. We explain the key algorithmic concepts behind each of these components below.

**Review of the FastMap algorithm**. FastMap[18] is a Data Mining algorithm that embeds complex objects—such as audio signals, seismograms, DNA sequences, electrocardiograms, or magnetic-resonance images—into a $K$-dimensional Euclidean space, for a user-specified value of $K$ and a user-supplied function $\mathcal{D}$ that quantifies the distance, or dissimilarity, between pairs of objects. The Euclidean distance between any two objects in the embedding produced by FastMap approximates the domain-specific distance between them. Therefore, similar objects, as quantified by $\mathcal{D}$, map to nearby points in Euclidean space; whereas dissimilar objects map to distant points. Although FastMap preserves $O(N^2)$ pairwise distances between $N$ objects, it generates the embedding in only $O(KN)$ time. Because of its efficiency, FastMap has already found numerous real-world applications, including in Data Mining[18], shortest-path computations[19], community detection and block modeling[20], and solving combinatorial optimization problems on graphs[21].

Below, we review the FastMap algorithm[18] and describe our minor modifications to it. These modifications suit the purposes of the downstream classification task. Our review of FastMap also serves completeness and the readers' convenience.

FastMap embeds a collection of complex objects in an artificially created Euclidean space that enables geometric interpretations, algebraic manipulations, and downstream application of ML algorithms. It gets as input a collection of complex objects $\mathcal{O}$ and a distance function $\mathcal{D}(\cdot, \cdot)$, where $\mathcal{D}(O_i, O_j)$ represents the domain-specific distance between objects $O_i, O_j \in \mathcal{O}$. It generates a Euclidean embedding that assigns a $K$-dimensional point $\mathbf{p}_i = \left( p_{i,1}, p_{i,2}, \dots, p_{i,K} \right) \in \mathbb{R}^K$ to each object $O_i$. A good Euclidean embedding is one in which the Euclidean distance $\| \mathbf{p}_i - \mathbf{p}_j \|_2 \equiv \sqrt{\sum_{n=1}^{K} (p_{i,n} - p_{j,n})^2}$ between any two points $\mathbf{p}_i$ and $\mathbf{p}_j$ closely approximates $\mathcal{D}(O_i, O_j)$.

FastMap creates a $K$-dimensional Euclidean embedding of the complex objects in $\mathcal{O}$, for a user-specified value of $K$. In the first iteration, FastMap heuristically identifies the farthest pair of objects $O_a$ and $O_b$ in linear time. Once $O_a$ and $O_b$ are determined, every other object $O_i$ defines a triangle with sides of lengths $d_{ai} = \mathcal{D}(O_a, O_i)$, $d_{ab} = \mathcal{D}(O_a, O_b)$, and $d_{ib} = \mathcal{D}(O_i, O_b)$ (Fig. 7). The side lengths of the triangle define its entire geometry, and the projection of $O_i$ onto the line $\overline{O_a O_b}$ is given by

$$x_i = \left( d_{ai}^2 + d_{ab}^2 - d_{ib}^2 \right) / (2 d_{ab}). \tag{2}$$

FastMap sets the first coordinate of $\mathbf{p}_i$, the embedding of $O_i$, equal to $x_i$. In the subsequent $K - 1$ iterations, FastMap computes the remaining $K - 1$ coordinates of each object following the same procedure; however, the distance function is adapted for each iteration. In the first iteration, the coordinates of $O_a$ and $O_b$ are 0 and $d_{ab}$, respectively. Because these coordinates perfectly encode the true distance between $O_a$ and $O_b$, the rest of $\mathbf{p}_a$ and $\mathbf{p}_b$'s coordinates should be identical for all subsequent iterations. Intuitively, this means that the second iteration should mimic the first one on a hyperplane that is perpendicular to the line $\overline{O_a O_b}$ (Fig. 8). Although the hyperplane is never explicitly constructed, it conceptually implies that the distance function for the second iteration should be changed for all $i$ and $j$ in the following way:

$$\mathcal{D}_{new}(O_i', O_j')^2 = \mathcal{D}(O_i, O_j)^2 - (x_i - x_j)^2, \tag{3}$$

in which $O_i'$ and $O_j'$ are the projections of $O_i$ and $O_j$, respectively, onto this hyperplane, $x_i$ and $x_j$ are the coordinates of $O_i$ and $O_j$ from the previous iteration, respectively, and $D_{new}(\cdot, \cdot)$ is the new distance function. The distance function is recursively updated according to Eq. (3) at the beginning of each of the $K - 1$ iterations that follow the first.

**Selecting reference objects**. As described before, in each of the $K$ iterations, FastMap heuristically finds the farthest pair of objects according to the distance function defined for that iteration. These objects are called pivots and are stored as reference objects. There are exactly $2K$ reference objects in our implementation because we prohibit any object from serving as a reference object more than once; however, this restriction is not strictly necessary. Technically, finding the farthest

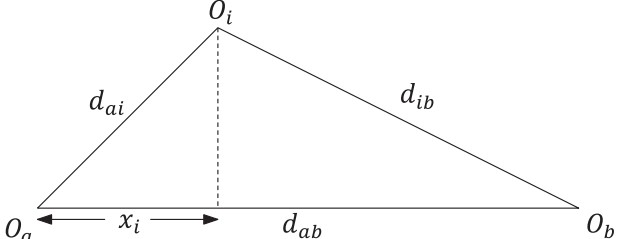

**Fig. 7 "Cosine law" employed by FastMap.** Shows the "cosine law" projection in a triangle.

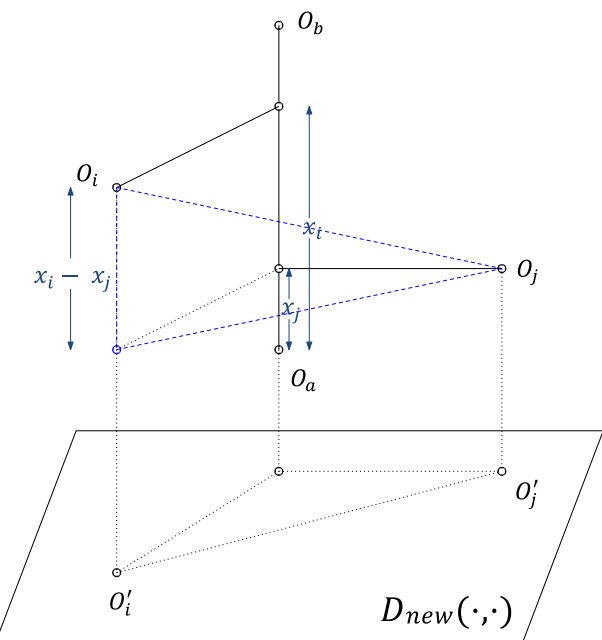

**Fig. 8 Hyperplane projection employed conceptually by FastMap.** Shows the projection onto a hyperplane that is perpendicular to $\overline{O_a O_b}$.

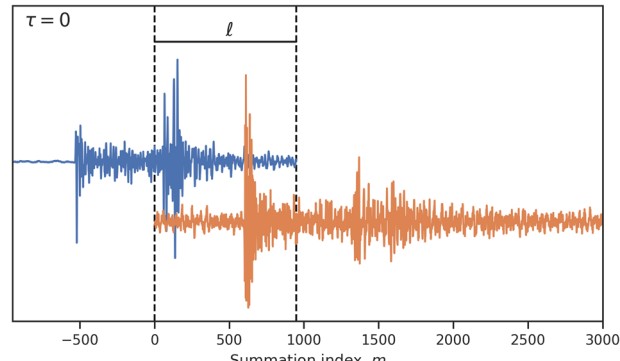

**Fig. 9 Schematic illustration of some quantities in Eq. (4).** Shows the alignment of a longer waveform in orange with a shorter waveform in blue at the first step in the normalized cross-correlation procedure ($\tau = 0$). The quantity $\ell$ measures about half the length of the shorter waveform.

pair of objects in any iteration takes $O(N^2)$ time. However, FastMap uses a linear-time "pivot changing" heuristic[18] to efficiently and effectively identify a pair of objects $O_a$ and $O_b$ that is very often the farthest pair. It does this by initially choosing a random object $O_b$ and then choosing $O_a$ to be the farthest object away from $O_b$. It then reassigns $O_b$ to be the farthest object away from $O_a$.

In our adaptation of FastMap as a component of FastMapSVM, we require the farthest pair of objects $O_a$ and $O_b$ in each iteration to be of opposite classes. This maximizes the discriminatory power of the downstream SVM classifier. We achieve this requirement by implementing a minor modification of the pivot changing heuristic: We initially choose a random object $O_b$. We then choose $O_a$ to be the farthest object away from $O_b$ and of the opposite class. We finally reassign $O_b$ to be the farthest object away from $O_a$ and of the opposite class. In each iteration, all previously used reference objects are excluded from consideration when selecting the pivots.

For a test object not seen before, its Euclidean coordinates in the $K$-dimensional embedding can be computed by using only its distances to the reference objects. This is based on the reasonable assumption that the new test object would not preclude the stored reference objects from being pivots if the $K$-dimensional Euclidean embedding was recomputed along with the new test object. In any case, the assumption is not strictly required since the stored reference objects are close to being the farthest pairs.

**Choosing the distance function $\mathcal{D}$.** The distance function should yield non-negative values for all pairs of objects and 0 for identical objects. We can use a variety of distance functions, such as the Wasserstein distance, the Jensen-Shannon divergence, or the Kullback-Leibler divergence. We can also use more domain-specific knowledge in the distance function, as described below.

In the Earthquake Science community, the normalized cross-correlation operator, denoted here by $\star$, is popularly used to measure similarity between two waveforms. For two zero-mean, single-component seismograms $O_i$ and $O_j$ with lengths $n_i$ and $n_j$, respectively, and starting with index 0, the normalized cross-correlation is defined with respect to a lag $\tau$ as follows:

$$(O_i \star O_j)[\tau] \triangleq \frac{1}{\sigma_i \sigma_j} \sum_{m=0}^{n_i-1} O_i[m] \widehat{O}_j[m + \ell - \tau], \quad (4)$$

in which, without loss of generality, we assume that $n_i \geq n_j$. $\sigma_i$ and $\sigma_j$ are the standard deviations of $O_i$ and $O_j$, respectively. Moreover, $\ell$ and $\widehat{O}_j$ are defined as follows:

$$\ell \triangleq \frac{n_j - n_j \,(\mathrm{mod}\,2)}{2} - \left(n_i \,(\mathrm{mod}\,2)\right)\left(1 - n_j \,(\mathrm{mod}\,2)\right) \quad (5)$$

and

$$\widehat{O}_j[m] \triangleq \begin{cases} O_j[m] & \text{if } 0 \leq m < n_j \\ 0 & \text{otherwise} \end{cases}. \quad (6)$$

The quantity $\ell$ in Eq. (5) is defined as a subtraction. The first term is approximately half of $n_j$. The second term is 0 or 1 depending on whether $n_i$ and $n_j$ are odd or even. Therefore, $\ell$ measures about half the length of the shorter waveform and ensures at

least 50 % overlap between the two waveforms for computing the normalized cross-correlation at any $\tau$ (Fig. 9).

Equipped with this knowledge, we first define the following distance function that is appropriate for waveforms with a single component:

$$\mathcal{D}(O_i, O_j) \triangleq 1 - \max_{0 \leq \tau \leq n_i - 1} \left| (O_i \star O_j)[\tau] \right|. \quad (7)$$

Based on this, we define the following distance function that is appropriate for waveforms with $L$ components:

$$\mathcal{D}(O_i, O_j) \triangleq 1 - \frac{1}{L} \max_{0 \leq \tau \leq n_i - 1} \left| \sum_{l=1}^{L} (O_i^l \star O_j^l)[\tau] \right|. \quad (8)$$

Here, each component $O_i^l$ of $O_i$, or $O_j^l$ of $O_j$, is a channel representing a one-dimensional data stream. A channel is associated with a single standalone sensor or a single sensor in a multi-sensor array.

We use the distance function defined in Eq. (8) with $L = 3$ for 3C seismograms. Our choice is motivated by the extensive use of similar equations in Earthquake Science to detect earthquakes using matched filters[13–16]. We will investigate other distance functions in future work.

**Enabling SVMs and Kernel methods.** SVMs are particularly good for classification tasks. When combined with kernel functions, they recognize and represent complex nonlinear classification boundaries very elegantly[22]. Moreover, soft-margin SVMs with kernel functions[23] can be used to recognize both outliers and inherent nonlinearities in the data. While the SVM machinery is very effective, it requires the objects in the classification task to be represented as points in a Euclidean space. Often, it is very difficult to represent complex objects like seismograms as precise geometric points without introducing inaccuracy or losing domain-specific representational features. In such cases, NNs have been more effective than SVMs. FastMapSVM revitalizes the SVM technology for classifying complex objects by leveraging the following observation: Although it may be hard to precisely describe complex objects as geometric points, it is often relatively easy to precisely compute the distance between any two of them. FastMapSVM uses the distance function to construct a low-dimensional Euclidean embedding of the objects. It then invokes the full power of SVMs. The low-dimensional Euclidean embedding also facilitates a perspicuous visualization of the classification boundaries.

**Implementing FastMapSVM.** We have implemented FastMapSVM and have made it publicly accessible in a Python package available at: https://github.com/malcolmw/FastMapSVM. Our package uses the `cupy` package[24] to compute the distance function in batches on a GPU. Our code also runs on CPU using Python's built-in `multiprocessing` module to accelerate distance function executions. FastMapSVM requires as input (1) the labeled train data set, (2) the distance function, (3) the dimensionality of the Euclidean embedding, and (4) a location to store the resulting trained model. We use the `thundersvm` package[25] to accelerate SVM computations on GPUs.

**Training deep learning models.** To train the EQTransformer and CRED models, we used 80% of the train data set to adjust the model weights via back propagation and reserved the other 20% for model validation. We trained the models from scratch using the SeisBench[10] framework for a maximum of 128 epochs using the Adam optimizer[26] with a learning rate of 0.001, and we stopped training if the validation loss failed to decrease for 8 consecutive epochs. Train labels are defined by a boxcar equal to 0 everywhere except between $t_P$ and $t_P + 1.4 \cdot (t_S - t_P)$, in which it is equal to 1, where $t_P$ and $t_S$ are the P- and S-wave arrival times registered

for a single event in STEAD. We train both models using the binary cross-entropy loss function:

$$\mathcal{L}_\theta(y, \widehat{y}) = -\frac{1}{N}\sum_{i=1}^{N} y_i \cdot \log(\widehat{y}_i) + (1 - y_i) \cdot \log(1 - \widehat{y}_i), \qquad (9)$$

where $y_i$ is the label on the $i^{th}$ train sample and $\widehat{y}_i$ is the predicted probability of the sample being labeled $y_i$. Note that the loss function we use to train EQTransformer is insensitive to the model outputs for identifying phases; we train it to only detect earthquakes.

## Data availability

STEAD data are publicly available at: https://github.com/smousavi05/STEAD. Ridgecrest data are publicly available at: https://scedc.caltech.edu.

## Code availability

Our implementation of FastMapSVM is publicly available at: https://github.com/malcolmw/FastMapSVM.

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

## Acknowledgements

This work at the University of Southern California is supported by DARPA under grant number HR001120C0157 and by NSF under grant number 2112533. The views, opinions, and/or findings expressed are those of the author(s) and should not be interpreted as representing the official views or policies of the sponsoring organizations, agencies, or the U.S. Government. We thank Mostafa Mousavi for impartial and insightful discussion during the revision of this manuscript.

## Author contributions

M.W. and T.K.S.K. conceived the general concept of combining FastMap with SVMs and kernel methods for classification of complex objects, independent of prior work[4]. M.W. also refined the concept in the Earthquake Science domain, implemented the FastMapSVM method presented here, conducted the experiments, and drafted the manuscript. K.S. and A.L. conducted various experiments using FastMapSVM in support of those presented here. N.N. and T.K.S.K. provided critical guidance and oversight to the project. A.L., T.K.S.K., N.N., and K.S. contributed significantly to manuscript revision.

## Competing interests

The authors declare no competing interests.
