## [Peer Review File · Communications Engineering]

Reviewer #1 (Remarks to the Author):

The authors have discussed the FastSVM-based model in the context of classifying seismograms and showcase its performance in detecting and identifying earthquakes using the time histories of ground motion waveforms. The paper is well-written, presents exciting results, and follows the body of work conducted in Engineering Seismology and Earthquake Science. However, there are several significant queries that the reviewer believes need to be addressed in the paper before consideration for publication.

- 1) It is not clear what is the emphasis of this study. Is it the FastSVM algorithm (which I don't think it can be, as the algorithm by itself is not highly novel and has already been similarly proposed), or is it the seismological application of FastSVM for the classification of complex objects showcasing the power of algorithm?

The way the paper is formatted, the case seems to be the latter, but the abstract and title deviate towards the former. For example, the abstract only contains two lines about the application of the algorithm to earthquakes, while the complete results and discussions are primarily based on the seismological results. If the authors' focus is on the former, they should discuss the results in general applications/framework and provide a smaller section of earthquake application as a case study; In contrast, if the focus is on the latter, they should give more details about the application and problem statement in the abstract as well as the title. Hence this mix-match approach can confuse the readers, and the authors should revise the manuscript accordingly.

- 2) The "Introduction" section is too long and provides redundant details about the algorithm. For example, the four points mentioned in paragraphs 3,4,5 and 6 can be combined into one paragraph, and details can be explained in the "Methods" section.
- 3) The critical contributions mentioned in the manuscript are pretty misleading. The authors should separate their contributions in terms of the application of the algorithm to the seismological problem and the general novelties of the algorithm itself. The manuscript mainly entails the algorithm's robustness in terms of seismological application. Hence, claims about algorithms being fast, interpretable, etc., should be refrained. The authors shouldn't be claiming novelties about the algorithm that are already published in other studies. E.g., contributions #1 and #3 look identical and provide the same message in this context. Again, it can be a confusion caused by the mix-match narrative provided in the paper (as mentioned in comment #1)
- 4) Also, there are some "exaggerating" statements, e.g., Line 93-94: "We propose FastMapSVM as an advantageous alternative to other ML algorithms for general classification tasks." The authors say the algorithm is an "advantageous alternative" for

"general" classification, but they showcase the results only for seismological purposes, which is not "general" but rather a specific application. Hence, the reviewer believes that the authors should be careful in their claims and soften the tone.

- 5) The datasets are not adequately explored. The authors should provide more statistical details about the magnitude, rupture distances, peak ground acceleration/velocities/displacements, etc. This will allow the readers to understand better the robustness of the approach proposed by the authors.
- 6) In ML, the dataset is generally split into training and testing sets, and the model is trained using the Train set while the model does not observe the Test set during the training. However, in this manuscript, things get pretty confusing about how the datasets are separated in each section. E.g., in line 117, it is mentioned that they use 1.266×10^6 motions, but in line 125, they say only 538 are used to assess the model. And then again, in lines 156-163, they specify different numbers and percentages. The dataset numbers get confusing throughout the manuscript, and it's hard to keep track of these numbers. Due to the data-driven nature of ML models, the datasets used for training and testing must be exhaustive and have no biases. And same should be reflected through proper sampling and plots/analysis.
- 7) Furthermore, it is unclear how the earthquakes and noise data is selected, is it conducted randomly or based on some principles? Have the authors ensured that the datasets used for training and testing are not biased? There seems to be no test conducted for assessing such biases. Such tests are necessary at least to check the model's performance for various magnitude, distances, ground shaking levels, etc. Also, the sampling for training and testing must be conducted while considering such metrics.
- 8) In lines 170-173, the authors describe how the EQTransformer model is trained to detect earthquakes and identify phases simultaneously, while such an approach is not possible for FastMapSVM. In the reviewer's opinion, this comment is crucial, and such multi-output nature and maintenance of inter-correlations make NN models more appealing. Although two different end-to-end models can be used for the same purpose, such methods need explicit techniques to combine the outputs of the various models. For example, if the EQ is not detected, no inputs should lead to any phase identification (since there is no EQ, to begin with). While I understand this can be done trivially by putting an "if-else" condition in the two end-to-end models, the implicit maintenance of such relations by the NN models like Transformers is advantageous. The authors should reflect on such issues and sufficiently discuss their approach's other limitations, such as the assumption of linearity, homogeneity, etc.
- 9) The conclusions in lines 182 to 193 are a bit limited and confusing. The fact that the precision doesn't increase with the amount of training data while recall tends to increase significantly with higher data doesn't seem to be associated with what the authors have

mentioned in lines 184-185. The main difference between recall and precision is the denominator, where recall has FN while precision has FP. Hence, if precision doesn't change with amount of data while recall goes from low to high levels, the FN tends to decrease with increased data and discussion along that line. The reviewer is not sure how the authors arrived at lines 184-193. Please revisit these lines and provide more details.

- 10) Fig 2; the legend should be kept outside Figure 2a to present it as a general legend for both sub-figures.
- 11) Fig 4 and discussion around it: Firstly, it is unclear what time histories represent? Are they ground acceleration, velocities, or displacements? And what are their units? Adding a Gaussian noise with standard deviations of 0 to 10 is not conclusive unless the units of the time series are precise. For example, the noise of 10 sigmas is huge for acceleration values in units of g but may not be so much for acceleration in units of in/s^2 or cm/s^2 . Also, it looks weird that with an increase in noise, the amplitude for the different time histories remains quite identical while at the same time, the pulse at 6-second is observed to be gone with an increase in noise levels. This makes the figure a bit misleading and should be made per the scale with proper mention of the units.
- 12) The authors keep claiming that the FastSVM is more interpretable than NNs throughout the manuscript; however, I would like to suggest that the authors lower the tone a bit as the interpretation is not so clear. For example, in Fig 6, it is not straightforward what x_1 and x_0 Euclidean variables represent physically. The reviewer agrees that the algorithm is not as black-box natured as NNs, but it isn't a white box model like linear regression. And there is generally a trade-off between accuracy and interpretability.
- 13) The discussions on Pages 18-20 are identical to the introduction section with significant overlap. The authors should try to analyze the consequences of the results that they have observed and provide critical comments on them, rather than repeating the "superiority" of the algorithm.

Reviewer #2 (Remarks to the Author):

The authors of this paper combine two well-known methods in machine learning, FastMap and SVM, to classify seismic data using two benchmark datasets. As a distance measure, they use cross-correlation, and the method is called FastMapSVM. The results are then compared with deep learning models in the community of seismology and claim that FastMapSVM is superior. This paper is well written and well explained, supporting it with codes and many plots for clarification. Considering the results, this method appears to be a better choice than current Deep Learning models in the seismology community, since it requires less training effort with almost the same accuracy.

Upon investigating the data set provided with the Ridgecrest name in the GitHub link, I realized it would be very simple to classify noise and Eqs. Using Deep Learning models with millions of free parameters for this simple task seems ridiculous to me. In my opinion, even the model presented here is overcomplicated. Using a simple average distance to the training data, I could cluster the test data much more effectively. Using this simple average distance clustering, attached is a plot comparable to Figure 6 in the manuscript.

When the ratio between the average distance to EQ and the average distance to noise is greater than one, waveforms are classified as EQ, while if it is smaller than one, they are classified as Noise. Since this is so intuitive and effective, I believe the only waveform miscategorized is actually EQ and it has been mislabeled.

In some cases, the signal from EQs is so weak that it is below the noise level and can be mislabeled. The program runs in a few seconds on a desktop computer with no hyperparameters, and it is just a few lines of code.

In light of this simple test, I believe the paper might not be appropriate for publication in this journal, and I strongly advise the authors to submit it to a seismology journal, since I believe it is true that this method is better than those Deep Learning models in the seismology community. Computer science and signal processing communities have extensively studied time series classification for many years. Methods for this task should be compared to State-of-the-Art methods in the computer science and signal processing community (e.g., A, B) and tested on general and more complex data sets such as the UCR time series archive (C).

The method for detecting phase arrivals is also very specialized for seismology applications. This is yet another reason to publish it in a seismology journal.

A) Ismail Fawaz, Hassan, Germain Forestier, Jonathan Weber, Lhassane Idoumghar, and Pierre-Alain Muller. "Deep learning for time series classification: a review." *Data mining and knowledge discovery* 33, no. 4 (2019): 917-963.

B) Tong, Yuerong, Jingyi Liu, Lina Yu, Liping Zhang, Linjun Sun, Weijun Li, Xin Ning, Jian Xu, Hong Qin, and Qiang Cai. "Technology investigation on time series classification and prediction." *PeerJ Computer Science* 8 (2022): e982.

C) Dau, Hoang Anh, Anthony Bagnall, Kaveh Kamgar, Chin-Chia Michael Yeh, Yan Zhu, Shaghayegh

Gharghabi, Chotirat Ann Ratanamahatana, and Eamonn Keogh. "The UCR time series archive." *IEEE/CAA Journal of Automatica Sinica* 6, no. 6 (2019): 1293-1305.

Nader Shakibay Senobari

Reviewer #3 (Remarks to the Author):

In this work, the authors developed the FastMapSVM method, which combined two algorithms using FastMap to map data in an Euclidean feature space and using SVM to conduct classification in the transformed feature space. They applied the method to classifying earthquake signals from background noise and demonstrated that the detection performance is similar to state-of-the-art deep learning models. However, the advantages of FastMapSVM are clearly demonstrated. I think this work still needs more rigorously designed experiments to demonstrate the claimed advantages compared with current algorithms before publishing in a high impact journal like *Nature*. Because many researchers would want to test this method if it is an advantageous alternative to Neural Networks

Comments:

1. Advantage one: Interpretable.

I do not fully understand how the authors define interpretable. Are the two axes in Fig. 6 interpretable? What are the physical meanings of these two axes?

If the authors define "interpretable" as using these mapped features, you can also use these features as the inputs of a neural network model. Or you can just use an auto-encoder to learn some latent features without FastMap.

If the authors define "interpretable" as drawing the classification boundaries. You can also scan the latent feature space learned by a neural network to produce a similar figure as Fig. 6.

From the Results section, I do not find that the authors explained clearly how to interpret the model, analyze which parameter is controlling the prediction, or change the parameter to improve the model. More analysis is needed to explain the advantage of "Interpretable"

2. Advantage two: training data size

The authors used only ~1.295 % (i.e., 16,384) of the STEAD waveforms to train FastMapSVM and achieved comparable results as neural network models. But this comparison is not accurate, because these deep learning models do not have to use the whole $1.2e6$ training samples to achieve the reported performance. A rigorous comparison is to train a neural network using the same training samples as FastMapSVM, and compare the increase of model performance with the increase of training samples as Fig. 2a.

The STEAD dataset is known to be a bit "too clean", most machine learning models can achieve a similar performance using a small set of training samples.

3. Advantage three: training time

The training time is an advantage because FastMapSVM is based on SVM, which is known to be much faster than neural networks in training. On the other hand, the prediction speed of neural network models is very fast. It is necessary to add the comparisons of prediction speeds too.

4. Why do you use only one station (TA.109C) for phase arrival identification? Is the similarity between phases an important factor for the performance of the model?

5. "The recall remains at or close to 1 irrespective of the amplitude of the noisy perturbations" What threshold do you use?

6. "During this time period, the SCEDC earthquake catalog reports no earthquakes within 100 km of CI.CLC; however, FastMapSVM identifies 19 windows with earthquakes. Of these, 9 contain clear earthquake signals with easily discernible P- and S-wave arrivals" This is not an accurate comparison, because FastMapSVM uses only one station. The SCEDC catalog is generated after associating multiple stations. Their phase detector may have detected these arrivals, but they are too weak to be associated by multiple station. You would need to run FastMapSVM on multiple stations to compare with the SCEDC catalog.

8. The Ridgecrest earthquake is a good test case since there are several catalogs generated by different methods. Instead of running on one station, it is better to use multiple stations nearby and compare the generated catalogs with other catalogs. This can be a very convincing result to demonstrate the performance of this method.

9. Some quantitative experiments can help the audience to understand the improvements of FastMapSVM over the SupFM-SVM method of Ban et al.

10. Eq. 4: Could you explain what is "l" doing here?

11. Eq. 3 - Eq. 6: Is this part same as conventional template matching processing? If so, I do not understand why this algorithm is very fast? For earthquake detection on continuous data, template matching is much more computationally intensive than neural networks. It would be helpful to compare the prediction speed on continuous data.

Review #1

Review of: "FastMapSVM: Classifying Complex Objects Using the FastMap Algorithm and Support-Vector Machines"

The authors have discussed the FastSVM-based model in the context of classifying seismograms and showcase its performance in detecting and identifying earthquakes using the time histories of ground motion waveforms. The paper is well-written, presents exciting results, and follows the body of work conducted in Engineering Seismology and Earthquake Science. However, there are several significant queries that the reviewer believes need to be addressed in the paper before consideration for publication.

Dear Reviewer,

We thank you for taking time to review our manuscript and appreciate your constructive criticisms. We have reviewed each of your comments carefully and responded to them one by one below. We hope that you will find that we have addressed your concerns clearly and completely.

1) It is not clear what is the emphasis of this study. Is it the FastSVM algorithm (which I don't think it can be, as the algorithm by itself is not highly novel and has already been similarly proposed), or is it the seismological application of FastSVM for the classification of complex objects showcasing the power of algorithm?

The emphasis of the revised paper is along the lines of the latter of your suggestions: We use applications in the seismogram domain to demonstrate that FastMapSVM offers a lightweight alternative to state-of-the-art algorithms (e.g., neural networks (NNs)) for classifying complex objects when training data or time is limited. We have revised the manuscript throughout to reflect this emphasis and make it clearer when we are extrapolating our results to other domains.

The way the paper is formatted, the case seems to be the latter, but the abstract and title deviate towards the former. For example, the abstract only contains two lines about the application of the algorithm to earthquakes, while the complete results and discussions are primarily based on the seismological results. If the authors' focus is on the former, they should discuss the results in general applications/framework and provide a smaller section of earthquake application as a case study; In contrast, if the focus is on the latter, they should give more details about the application and problem statement in the abstract as well as the title. Hence this mix-match approach can confuse the readers, and the authors should revise the manuscript accordingly.

Thank you for pointing this out. We agree that the apparent focus on the seismological application of FastMapSVM is at odds with the emphasis conveyed by the title and

abstract of the original manuscript. We have revised the title and abstract to reflect the emphasis of the revised paper as discussed in our response to the comment above. Our motivation is to a) demonstrate that FastMapSVM outperforms state-of-the-art models in the seismogram domain when training data or time is limited, and b) use these results to motivate FastMapSVM as a potentially advantageous alternative to NNs in other domains. To further motivate FastMapSVM in other domains, we include additional results as Supplementary Material from a second application in which we classify images of hand-written digits in the MNIST database (LeCun et al., 2010) with >98% accuracy using FastMapSVM.

2)The "Introduction" section is too long and provides redundant details about the algorithm. For example, the four points mentioned in paragraphs 3,4,5 and 6 can be combined into one paragraph, and details can be explained in the "Methods" section.

Thank you for this feedback. We have revised the introduction to make it shorter by removing non-essential and redundant information. We have integrated key points from paragraphs 3, 4, 5, and 6 into the Discussion section.

3) The critical contributions mentioned in the manuscript are pretty misleading. The authors should separate their contributions in terms of the application of the algorithm to the seismological problem and the general novelties of the algorithm itself. The manuscript mainly entails the algorithm's robustness in terms of seismological application. Hence, claims about algorithms being fast, interpretable, etc., should be refrained. The authors shouldn't be claiming novelties about the algorithm that are already published in other studies. E.g., contributions #1 and #3 look identical and provide the same message in this context. Again, it can be a confusion caused by the mix-match narrative provided in the paper (as mentioned in comment #1)

Thank you for this feedback. We have carefully revised our Main Contributions and paid close attention to a) make clear distinction between our results in the seismogram domain and our extrapolation of those results to other domains, and b) not claim novelties that are already published elsewhere.

4) Also, there are some "exaggerating" statements, e.g., Line 93-94: "We propose FastMapSVM as an advantageous alternative to other ML algorithms for general classification tasks." The authors say the algorithm is an "advantageous alternative" fo"general" classification, but they showcase the results only for seismological purposes, which is not "general" but rather a specific application. Hence, the reviewer believes that the authors should be careful in their claims and soften the tone.

Thank you for pointing this out. We agree that the claim was too general, not sufficiently supported by the results, and worded too strongly. The term *general* is nebulous and may be misconstrued as implying that FastMapSVM is advantageous for *any* or *every* classification task, which is not the case. We rephrase this statement to specify that

FastMapSVM is a *lightweight* alternative to NN models for classification tasks *in which training data or time is limited and a suitable distance function can be defined*. We have also revised the manuscript to say that our results in the seismogram domain motivate FastMapSVM as a *potentially* advantageous alternative to NNs in other domains.

5) The datasets are not adequately explored. The authors should provide more statistical details about the magnitude, rupture distances, peak ground acceleration/velocities/displacements, etc. This will allow the readers to understand better the robustness of the approach proposed by the authors.

Thank you for pointing out this lack of important detail. We have revised our discussion of the data sets to include their key attributes and added supplementary figures summarizing the data distributions.

6) In ML, the dataset is generally split into training and testing sets, and the model is trained using the Train set while the model does not observe the Test set during the training. However, in this manuscript, things get pretty confusing about how the datasets are separated in each section. E.g., in line 117, it is mentioned that they use 1.266x10⁶ motions, but in line 125, they say only 538 are used to assess the model. And then again, in lines 156-163, they specify different numbers and percentages. The dataset numbers get confusing throughout the manuscript, and it's hard to keep track of these numbers. Due to the data-driven nature of ML models, the datasets used for training and testing must be exhaustive and have no biases. And same should be reflected through proper sampling and plots/analysis.

Thank you for pointing out these confusing aspects of the manuscript. We have carefully revised our description of the data sets and their various subsets, making sure we specify how each data set is constructed and what it is used for. Furthermore, we have revised our approach to selecting data (partly in response to a comment from another reviewer) to mitigate potential biases. We believe that you will find this aspect of the revised manuscript easier to follow.

7) Furthermore, it is unclear how the earthquakes and noise data is selected, is it conducted randomly or based on some principles? Have the authors ensured that the datasets used for training and testing are not biased? There seems to be no test conducted for assessing such biases. Such tests are necessary at least to check the model's performance for various magnitude, distances, ground shaking levels, etc. Also, the sampling for training and testing must be conducted while considering such metrics.

Thank you for raising this point. We have revised our approach to selecting data and reflected this in our descriptions of the data and sampling procedures.

You will see in the revised Data section that we select random subsets from STEAD in a manner intended to broaden the data distribution and mitigate biases.

The Ridgecrest data set was selected randomly from the analyst-reviewed earthquake catalog from the Southern California Seismic Network. This catalog is itself

biased towards larger magnitude earthquakes, as there are too many small earthquakes to be reviewed by analysts. Our Ridgecrest data set thus reflects this bias; however, we believe this is not a problem because of the application of the data set.

The first application of the Ridgecrest data set is in testing against noisy perturbations. We intentionally design a baseline test case for which FastMapSVM can obtain nearly perfect accuracy, and then subsequently perturb the input data with random noise to observe the model's performance as a function of the amount of added noise. Any biases in the data set that favor strong model performance in the baseline test case are counteracted by added noise in the subsequent iterations.

The second application of the Ridgecrest data set is in a "real-world" deployment scenario in which a set of analyst-reviewed earthquake seismograms are used to detect new events. Despite any biases that may result from the analyst review procedure in such a real-world use case, FastMapSVM still outperforms the existing algorithms we compared it against.

In all cases, we draw simple random samples from our base data sets so that subsets have the same statistical properties of the data set from which they were drawn. Furthermore, we conduct multiple iterations of each test and present statistical properties of the results to further mitigate biases in individual test cases.

8) In lines 170-173, the authors describe how the EQTransformer model is trained to detect earthquakes and identify phases simultaneously, while such an approach is not possible for FastMapSVM. In the reviewer's opinion, this comment is crucial, and such multi-output nature and maintenance of inter-correlations make NN models more appealing. Although two different end-to-end models can be used for the same purpose, such methods need explicit techniques to combine the outputs of the various models. For example, if the EQ is not detected, no inputs should lead to any phase identification (since there is no EQ, to begin with). While I understand this can be done trivially by putting an "if-else" condition in the two end-to-end models, the implicit maintenance of such relations by the NN models like Transformers is advantageous. The authors should reflect on such issues and sufficiently discuss their approach's other limitations, such as the assumption of linearity, homogeneity, etc.

We agree that a single model that implicitly maintains logical consistency between multiple interrelated outputs is appealing; however, it is not the case that EQTransformer does anything fundamentally different from what can be done with FastMapSVM. EQTransformer effectively comprises three end-to-end models (one each for event detection, P-wave picking, and S-wave picking) with additional post-processing that combines their outputs in a logically consistent manner. The three separate end-to-end models do share a common encoder branch, but this does not maintain any notion of consistency between the three parallel, downstream decoders in any intuitive way. It would not be particularly difficult to implement an augmented FastMapSVM API that manages multiple internal models and applies post-processing to their outputs in a similar fashion to EQTransformer. In fact, this is what we do to obtain results for classifying images of hand-written digits included as Supplementary Material. We refrain from doing so in the seismogram domain, however, because we want to emphasize the

general characteristics of the algorithm, and doing so would entail specializing the algorithm for our particular seismological application. Note that the only component of the algorithm that is tailored to the seismogram domain is our definition of the distance function.

We have left out any discussion of this in the manuscript (and removed the statement that led to the Reviewer's comment in the first place) because it is somewhat irrelevant with respect to the revised results. In the revised manuscript, we train EQTransformer from scratch using a loss function that is insensitive to the output of the P- and S-wave picking branches of the model. In other words, we compare FastMapSVM against a version of EQTransformer trained to only perform event detection.

9) The conclusions in lines 182 to 193 are a bit limited and confusing. The fact that the precision doesn't increase with the amount of training data while recall tends to increase significantly with higher data doesn't seem to be associated with what the authors have mentioned in lines 184-185. The main difference between recall and precision is the denominator, where recall has FN while precision has FP. Hence, if precision doesn't change with amount of data while recall goes from low to high levels, the FN tends to decrease with increased data and discussion along that line. The reviewer is not sure how the authors arrived at lines 184-193. Please revisit these lines and provide more details.

Thank you for drawing attention to this section of the manuscript. We have revised it in a way that we believe makes it clearer. In the original manuscript we presented "binary" performance scores, which only represent results for the positive (earthquake) class. We report "macro" averages in the revised manuscript, which represent scores averaged over both positive and negative classes and are thus unbiased.

10) Fig 2; the legend should be kept outside Figure 2a to present it as a general legend for both sub-figures.

Thank you for this suggestion. The revised Figure 2 no longer has a shared legend, but we have made sure to keep the shared legend outside the subplots in revised Figure 1.

11) Fig 4 and discussion around it: Firstly, it is unclear what time histories represent? Are they ground acceleration, velocities, or displacements? And what are their units? Adding a Gaussian noise with standard deviations of 0 to 10 is not conclusive unless the units of the time series are precise. For example, the noise of 10 sigmas is huge for acceleration values in units of g but may not be so much for acceleration in units of in/s^2 or cm/s^2 . Also, it looks weird that with an increase in noise, the amplitude for the different time histories remains quite identical while at the same time, the pulse at 6-second is observed to be gone with an increase in noise levels. This makes the figure a bit misleading and should be made per the scale with proper mention of the units.

We have added a statement to the Data subsection indicating that all seismograms used in this study are records of ground velocity. We have also indicated in the discussion of the relevant figure that test seismograms are self-normalized by their standard deviation before adding noise, which makes them unitless. Furthermore, we indicate that traces are again self-normalized for plotting, which is why the relative amplitudes might seem peculiar on the first look. We expect, however, that viewing plots of self-normalized seismograms will be familiar to the readers.

12) The authors keep claiming that the FastSVM is more interpretable than NNs throughout the manuscript; however, I would like to suggest that the authors lower the tone a bit as the interpretation is not so clear. For example, in Fig 6, it is not straightforward what x_1 and x_0 Euclidean variables represent physically. The reviewer agrees that the algorithm is not as black-box natured as NNs, but it isn't a white box model like linear regression. And there is generally a trade-off between accuracy and interpretability.

Thank you for this suggestion. We appreciate that the tone may be too strong in certain passages and have revised the manuscript throughout to soften it. We have also added discussion elaborating on this point to make it clearer what we mean by "interpretable."

The x_0 and x_1 Euclidean coordinates represent the relative similarity of a given object to the pivot objects for the 0th and 1st dimensions, respectively. In other words, a point with $x_0 > 0.5$ represents a seismogram that is more similar (in the sense defined by the user-supplied distance function) to the 0th pivot object from the earthquake class than the 0th pivot object from the noise class. A point with $x_0 < 0.5$ represents a seismogram that is more similar to the 0th noise pivot. Similar statements hold for the higher dimensions.

13) The discussions on Pages 18-20 are identical to the introduction section with significant overlap. The authors should try to analyze the consequences of the results that they have observed and provide critical comments on them, rather than repeating the "superiority" of the algorithm.

Thank you for pointing out this redundancy. We have revised the Discussion section significantly, paying attention to discussing the impact of our results. We have also removed the overlapping content from the Introduction and integrated most of it into the Discussion.

Review #2

Reviewer #2 (Remarks to the Author):

The authors of this paper combine two well-known methods in machine learning, FastMap and SVM, to classify seismic data using two benchmark datasets. As a distance measure, they use cross-correlation, and the method is called FastMapSVM. The results are then compared with deep learning models in the community of seismology and claim that FastMapSVM is superior.

This paper is well written and well explained, supporting it with codes and many plots for clarification. Considering the results, this method appears to be a better choice than current Deep Learning models in the seismology community, since it requires less training effort with almost the same accuracy.

Dear Dr. Senobari,

We thank you for reviewing our manuscript, and appreciate your comments. We have addressed each of them below and hope that you will find our responses satisfactory.

Upon investigating the data set provided with the Ridgecrest name in the GitHub link, I realized it would be very simple to classify noise and Eqs. Using Deep Learning models with millions of free parameters for this simple task seems ridiculous to me. In my opinion, even the model presented here is overcomplicated. Using a simple average distance to the training data, I could cluster the test data much more effectively. Using this simple average distance clustering, attached is a plot comparable to Figure 6 in the manuscript.

When the ratio between the average distance to EQ and the average distance to noise is greater than one, waveforms are classified as EQ, while if it is smaller than one, they are classified as Noise. Since this is so intuitive and effective, I believe the only waveform miscategorized is actually EQ and it has been mislabeled.

In some cases, the signal from EQs is so weak that it is below the noise level and can be mislabeled. The program runs in a few seconds on a desktop computer with no hyperparameters, and it is just a few lines of code.

In light of this simple test, I believe the paper might not be appropriate for publication in this journal, and I strongly advise the authors to submit it to a seismology journal, since I believe it is true that this method is better than those Deep Learning models in the seismology community. Computer science and signal processing communities have extensively studied time series classification for many years. Methods for this task should be compared to State-of-the-Art methods in the computer science and signal processing community (e.g., A, B) and tested on general and more complex data sets such as the UCR time series archive (C).

The method for detecting phase arrivals is also very specialized for seismology applications. This is yet another reason to publish it in a seismology journal.

A) Ismail Fawaz, Hassan, Germain Forestier, Jonathan Weber, Lhassane Idoumghar, and Pierre-Alain Muller. "Deep learning for time series classification: a review." *Data mining and knowledge discovery* 33, no. 4 (2019): 917-963.

B) Tong, Yuerong, Jingyi Liu, Lina Yu, Liping Zhang, Linjun Sun, Weijun Li, Xin Ning, Jian Xu, Hong Qin, and Qiang Cai. "Technology investigation on time series classification and prediction." *PeerJ Computer Science* 8 (2022): e982.

C) Dau, Hoang Anh, Anthony Bagnall, Kaveh Kamgar, Chin-Chia Michael Yeh, Yan Zhu, Shaghayegh Gharghabi, Chotirat Ann Ratanamahatana, and Eamonn Keogh. "The UCR time series archive." *IEEE/CAA Journal of Automatica Sinica* 6, no. 6 (2019): 1293-1305.

Thank you for this feedback. We have decided to submit a revised version of the manuscript to *Communications Engineering* after carefully reviewing and responding to each comment made by the Reviewers.

Although we agree that DL models and FastMapSVM are unnecessarily complicated for this simple test case, we believe that it is unfair for the following reasons:

- a) In our paper, we randomly select 8 earthquake signals and 8 noise signals from the train data set included in the GitHub link. It is unclear whether such sampling was conducted or all 256 instances in the train data set were used in the Reviewer's test above. Will the simple algorithm proposed by the Reviewer work as well with such a limited train data size?
- b) It is unclear whether the distance function in the algorithm proposed by the Reviewer is the same cross-correlation based distance function used in the manuscript or a simple p-norm. If the latter is the case, the reason that the simple method proposed above is able to classify seismograms so accurately is likely due in part to the fact that seismograms are already time aligned. We suspect that if they were not time aligned, the average distance metric would be less

diagnostic. Note that we randomly shift the test signals before classifying them to ensure that our results are independent of any *a priori* temporal alignment.

- c) We intentionally created this test case as a simple baseline for which we could attain high accuracy. We then add noisy perturbations to the test data and repeat the classification procedure to assess the model performance when significant amounts of noise are present. We suspect that the performance of the average distance metric proposed by the Reviewer will decrease rapidly in the presence of increasing noise, particularly when the temporal alignment issue above is accounted for.
- d) FastMapSVM only computes the distance function between test objects and two pivot objects per Euclidean dimension, whereas the algorithm proposed by the Reviewer computes the distance function between test objects and *each* train object. Thus, for each test object, the algorithm proposed by the Reviewer has $O(N)$ complexity with respect to train data size N , whereas FastMapSVM has $O(2K)$ complexity for Euclidean embedding dimensionality $K \ll N$.

Concerning the Reviewer's comment about the extensive prior work in the Computer Science community on time series classification, we certainly appreciate this point. A key emphasis of this manuscript, however, which we admittedly failed to communicate clearly in the original manuscript, is that FastMapSVM is not restricted to time series data. To make this point clearer, we have included additional results as Supplementary Material in which we classify images of hand-written digits from the MNIST database (LeCun et al., 2010). This is a 2-D image classification task for which we obtain >98% test accuracy using a simple distance function based on the cosine similarity of images. Conducting a rigorous comparison against state-of-the-art models for general time series and image classification tasks is beyond the scope of this paper. Our main intention is to motivate FastMapSVM as a lightweight alternative to those state-of-the-art methods.

Nader Shakibay Senobari

Review #3

Reviewer #3 (Remarks to the Author):

In this work, the authors developed the FastMapSVM method, which combined two algorithms using FastMap to map data in an Euclidean feature space and using SVM to conduct classification in the transformed feature space. They applied the method to classifying earthquake signals from background noise and demonstrated that the detection performance is similar to state-of-the-art deep learning models. However, the advantages of FastMapSVM are clearly demonstrated. I think this work still needs more rigorously designed experiments to demonstrate the claimed advantages compared with current algorithms before publishing in a

high impact journal like Nature. Because many researchers would want to test this method if it is an advantageous alternative to Neural Networks

Comments:

1. Advantage one: Interpretable.

I do not fully understand how the authors define interpretable. Are the two axes in Fig. 6 interpretable? What are the physical meaning of these two axes?

We never defined interpretable in the original manuscript. Thank you for pointing out this ambiguity. In the revised manuscript, we have defined interpretability as *the degree to which causal mappings between model inputs and outputs can be explained to humans* (e.g., Lechner et al. 2022). Indeed, Fig. 6 is interpretable: Each point represents a seismogram, and the Euclidean distance between any pair of points is approximately equal to the dissimilarity between the seismograms they represent as defined by Equation (7). A distant pair of points in Fig. 6 represents a pair of seismograms that are weakly correlated to one another, whereas a pair of closely spaced points represents seismograms that are strongly correlated with one another.

If the authors define “interpretable” as using these mapped features, you can also use these features as the inputs of a neural network model. Or you can just use auto-encoder to learn some latent features without FastMap.

It is true that the features extracted by FastMap could serve as input to a neural network model; however, the decision boundary determined by the SVM is optimal in the maximum-margin sense, whereas NNs offer no such guarantee of optimality. Domingos (2020) presents an interesting analysis indicating that NNs are approximately the same as kernel machines (e.g., SVMs). Thus, we don't expect NNs to yield superior results to our SVM approach, despite having increased training complexity.

Furthermore, although it is possible to use an auto-encoder in place of FastMap, it is likely that the auto-encoder will require significantly more time and data to train than FastMapSVM. This is largely because (1) auto-encoders consider each training instance individually and thus leverage $O(N)$ pieces of information from a database of size N , whereas FastMapSVM considers distances between pairs of objects, thus leveraging $O(N^2)$ pieces of information; and (2) FastMapSVM explicitly incorporates domain knowledge into the training procedure, whereas an auto-encoder would have to learn representational features entirely from the train data.

Domingos, P. (2020). Every Model Learned by Gradient Descent Is Approximately a Kernel Machine. *arXiv*. <https://doi.org/10.48550/arxiv.2012.00152>

If the authors define “interpretable” as drawing the classification boundaries. You can also scan the latent feature space learned by a neural network to produce a similar figure as Fig. 6.

The ability to draw the classification boundary is one facet of interpretability, but not the entirety of it. Interpretability is supported by the combination of (1) intuitive understanding of which object attributes lead to clustering in Euclidean space, and (2) the relatively simple geometric interpretation of the SVM decision-making process. It is possible to scan the latent feature space learned by a NN, but doing so is a relatively cumbersome way of interpreting causal mappings between model inputs and outputs, and the interpretation of latent features themselves is rarely obvious.

From the Results section, I do not find that the authors explained clearly how to interpret the model, analyze which parameter is controlling the prediction, or change the parameter to improve the model. More analysis is needed to explain the advantage of “Interpretable”

Thank you for pointing out this lack of clarity. We have revised the Results section to clarify what we mean by interpretability. Apart from the SVM hyperparameters, which are discussed extensively elsewhere, the key hyperparameters that influence the performance of FastMapSVM are the dimensionality of the Euclidean embedding and the size of the train data set. We dedicate a subsection of the Results section to addressing these aspects of model training.

2. Advantage two: training data size

The authors used only ~1.295 % (i.e., 16,384) of the STEAD waveforms to train FastMapSVM and achieved comparable results as neural network models. But this comparison is not accurate, because these deep learning models do not have to use the whole 1.2e6 training samples to achieve the reported performance. A rigorous comparison is to train a neural network using the same training samples as FastMapSVM, and compare the increase of model performance with the increase of training samples as Fig. 2a.

The STEAD dataset is known to be a bit “too clean”, most machine learning models can achieve a similar performance using a small set of training samples.

Thank you for mentioning this. We agree that the comparison was potentially misleading and have revised our results to illustrate the effect train data size has on the performance of each model, as you suggest. In these new results, we train CRED, EQTransformer, and FastMapSVM from scratch using successively larger, identical train data sets (train data size increases by a factor of two with each iteration) and test them on a single test data set. We report balanced F1, accuracy, precision, and recall performance scores, which indicate that FastMapSVM outperforms the NN models for small training data sizes and highlights the revised emphasis of the paper: *FastMapSVM is an advantageous, lightweight alternative to neural networks when the amount of training data or time is limited.*

3. Advantage three: training time

The training time is an advantage because FastMapSVM is based on SVM, which is known to be much faster than neural networks in training. On the other hand, the prediction speed of

neural network models is very fast. It is necessary to add the comparisons of prediction speeds too.

Thank you for this comment. We have added analysis of train and prediction speed to the paper. These results indicate that FastMapSVM can be significantly faster than both NN models for making predictions when the Euclidean embedding dimensionality is small. The prediction time for FastMapSVM is linearly proportional to the Euclidean embedding dimensionality, so the NNs will make faster predictions for large Euclidean embeddings, but strong performance can be achieved using small embeddings.

4. Why do you use only one station (TA.109C) for phase arrival identification? Is the similarity between phases an important factor for the performance of the model?

The waveform attributes that differentiate P-waves from S-waves are relatively subtle compared to those that differentiate noise from earthquakes. These subtleties are easily obfuscated by site effects, which are difficult to disentangle without a highly complicated model. To mitigate the difficulties imposed by site effects, we restrict our analysis to a single station.

5. “The recall remains at or close to 1 irrespective of the amplitude of the noisy perturbations” What threshold do you use?

We use a 0.5 threshold for classification with FastMapSVM in this analysis. We have added text indicating that we use 0.5 threshold unless otherwise specified. The only case that we use a different threshold is in the automatic scanning of Ridgecrest data, for which we set the threshold to 0.95 to ensure reliable detections.

6. “During this time period, the SCEDC earthquake catalog reports no earthquakes within 100 km of CI.CLC; however, FastMapSVM identifies 19 windows with earthquakes. Of these, 9 contain clear earthquake signals with easily discernible P- and S-wave arrivals” This is not an accurate comparison, because FastMapSVM uses only one station. The SCEDC catalog is generated after associating multiple stations. Their phase detector may have detected these arrivals, but they are too weak to be associated by multiple station. You would need to run FastMapSVM on multiple stations to compare with the SCEDC catalog.

We agree with your point: Comparing our results against the SCEDC catalog is unfair for the reason you mention. We have removed this statement and replaced it with a direct comparison to the pre-trained EQTransformer and CRED models. In this comparison we apply the EQTransformer and CRED models, which were pre-trained on large ($N > 1e6$), global data sets, to the same 10 minutes of data and compare the results. FastMapSVM convincingly outperforms EQTransformer and CRED on this test.

8. The Ridgecrest earthquake is a good test case since there are several catalogs generated by different methods. Instead of running on one station, it is better to use multiple stations nearby

and compare the generated catalogs with other catalogs. This can be a very convincing result to demonstrate the performance of this method.

We agree with your point; however, building a catalog for the Ridgecrest sequence is beyond the scope of this project.

9. Some quantitative experiments can help the audience to understand the improvements of FastMapSVM over the SupFM-SVM method of Ban et al.

FastMapSVM is conceptually identical to the SupFM-SVM method. What is novel about our work is that we present, to the best of our knowledge, the first application of the method to complex objects for which a combination of representation and classification learning is necessary. Our results motivate the application of the method to other complex domains.

10. Eq. 4: Could you explain what is “l” doing here?

We apologize for the unwieldy definition of “l” in Equation 4, but we wanted to communicate the details of our cross-correlation procedure as precisely as possible. The quantity “l” simply defines the minimum allowable amount of overlap between the signals being correlated. As defined “l” ensures that the output signal will be the same length as the longer of the two input signals. The various “(mod 2)” terms account for whether the input signals are even- or odd-length. We have added a figure to represent the role of “l” graphically, as well as some text describing its purpose.

11. Eq. 3 - Eq. 6: Is this part same as conventional template matching processing? If so, I do not understand why this algorithm is very fast? For earthquake detection on continuous data, template matching is much more computationally intensive than neural networks. It would be helpful to compare the prediction speed on continuous data.

Yes, Eq. 3 - Eq. 7 are highly similar to conventional template matching. Our understanding is that cross-correlation is not necessarily more computationally intensive than NNs, but rather that NNs benefit from highly optimized libraries (e.g., PyTorch, TensorFlow, and JAX) that leverage specialized hardware (e.g., GPUs and TPUs). In fact, the convolution operations that are essential to the NNs considered in this paper (EQTransformer relies on 1440 different convolutional kernels) have nearly identical computational complexity to cross-correlation. By implementing FastMapSVM for GPU processing, we see that it is indeed faster than NN models. Furthermore, after training, the distance function is only executed for each pair of test objects and pivot objects. There are a limited number of pivot objects (two per Euclidean dimension), so the number of cross-correlation operations executed is smaller than a conventional template matching procedure that uses many templates.

Reviewers' comments:

Reviewer #1 (Remarks to the Author):

The authors have satisfactorily addressed the comments raised by the reviewer.

Reviewer #2 (Remarks to the Author):

Dear editors and authors:

The resubmitted manuscript has been heavily revised in response to my concerns (and those of other reviewers as well) regarding the need for more supporting material for its generality claims. The ambiguities with respect that the FastMapSVM is a superior choice with respect to NN models in any data set now is removed from the manuscript. The paper now is more focused on the application of seismology and I don't see a major concern regarding the claims. The method itself is not very novel, but its application to seismic data is, and the materials and results in this article may have an important impact on seismology. Here is one major comment and some minor suggestions.

Major comment:

Among the main sections that readers and potential users of the document can find useful in the future is the automatic scanning section. There is a lack of quantitative results supported by evidence in this section. Words like "clear" or "ambiguous" are bothersome. By having some ground truth regarding the detections, it is possible to provide a confident comparison between different methods. An important part of earthquake detection is locating and associating earthquake signals. This provides secondary assurance about the detection. The author could use a single station among an array of stations to compare the method using only that station and validate the results by using nearby stations by locating candidate signals, enhancing the signals with stacking, etc., and providing the ground truth of the results. In addition, template matching could be used to match those events to the bigger events in the catalog. It is at least better to be matched with another earthquake rather than simply verifying visually. By brainstorming, I could suggest these two options. Other methods may be available to validate the results more confidently. This is important and can significantly improve the paper's sound.

Minor Comments:

1- It would be helpful to provide more specific information about the problem the article aims to solve if the primary goal is to learn an ML model quickly and with few data sets. What kinds of problems could be solved if we used this method instead of NN? What is the importance of learning models when there are few training data sets? As an example, how much resource would be saved if we used this method for the same task done by x et al?

2- I agree with other reviewers that being intuitive here does not seem to be defined very clearly. However, I think embedding the results into similarity distance space, plotting, and visualizing them would provide a greater sense of intuition than using NNs.

I believe intuition is most effective when we can look back and see what we missed and why we missed it. You can trace back the reason for detecting an event when you use template matching, for example. The probability of false detection can be calculated using a threshold for the cross-correlation coefficient.

3-There are other NN models available for earthquake detection. Even though it isn't possible to compare all of them, please explain why EQTransformer and CRED were chosen. It is important to explain why only these two NN models are being compared with the method. Here is an example of a paper that claims that its objective is to have good accuracy with minimal training.

O. M. Saad and Y. Chen, "Earthquake Detection and P-Wave Arrival Time Picking Using Capsule Neural Network," in IEEE Transactions on Geoscience and Remote Sensing, vol. 59, no. 7, pp. 6234-6243, July 2021, doi: 10.1109/TGRS.2020.3019520.

line 70:

Time: what does it mean? Time required for training? The structure of the NN model plays a role in this.

line 108: Use math notation, 2^n where n varies from k1 to k2.

Reviewer #3 (Remarks to the Author):

Thanks to the authors for considering my comments to improve the paper.

I agree with the suggestions of other reviewers that this paper may be more appropriate for a seismic journal. Because there are three limitations of this work: Firstly, the algorithmic innovation of the proposed machine learning method is limited. Secondly, the main contribution of the paper lies in applying two machine learning methods (FastMap and SVM) to classify earthquakes and noise. Thirdly, the claimed advantage of better performance when training data or time is limited is not significant, because there are actually many manual labels available for earthquake detection.

I have a few minor comments on the revision:

1. Fig. 1: Why does the training time not linearly increase with the training dataset size? And why are the test times different for EQT and CREAD? From this result, it seems that the advantage of FastMapSVM is not significant. The performance and speed of other simpler models like GPD and PhaseNet can be even better.

2. Line 402: "FastMapSVM, EQTransformer, and CREAD identify, , and 13 windows with earthquakes." Numbers are missing here.

3. Table 1: I feel the samples are too few to make a reliable conclusion here. It should be pretty easy to apply to more earthquakes, since the Ridgecrest earthquake sequence is very active.

Review #1

The authors have satisfactorily addressed the comments raised by the reviewer.

Thank you for taking the time to review our manuscript again.

Review #2

Dear editors and authors:

The resubmitted manuscript has been heavily revised in response to my concerns (and those of other reviewers as well) regarding the need for more supporting material for its generality claims. The ambiguities with respect that the FastMapSVM is a superior choice with respect to NN models in any data set now is removed from the manuscript. The paper now is more focused on the application of seismology and I don't see a major concern regarding the claims. The method itself is not very novel, but its application to seismic data is, and the materials and results in this article may have an important impact on seismology. Here is one major comment and some minor suggestions.

Thank you for taking the time to review our manuscript again. We appreciate your constructive feedback and have addressed each of your comments individually below. We hope that you find we have addressed your concerns thoroughly.

Major comment:

Among the main sections that readers and potential users of the document can find useful in the future is the automatic scanning section. There is a lack of quantitative results supported by evidence in this section. Words like "clear" or "ambiguous" are bothersome. By having some ground truth regarding the detections, it is possible to provide a confident comparison between different methods. An important part of earthquake detection is locating and associating earthquake signals. This provides secondary assurance about the detection. The author could use a single station among an array of stations to compare the method using only that station and validate the results by using nearby stations by locating candidate signals, enhancing the signals with stacking, etc., and providing the ground truth of the results. In addition, template matching could be used to match those events to the bigger events in the catalog. It is at least better to be matched with another earthquake rather than simply verifying visually. By brainstorming, I could suggest these two options. Other methods may be available to validate the results more confidently. This is important and can significantly improve the paper's sound.

Thank you for this comment. We agree that the paper will benefit from better quantifying this section. To this end, we have made two major revisions to this section: we (1) analyze an entire day of data instead of only ten minutes, and (2) replaced subjective classifications (i.e., “clear”, “low-SNR”, “ambiguous”, and “false”) with measurements of maximum SNR and cross-correlation coefficient (measured against all 256 template seismograms used to train FastMapSVM) for each detection. We compare features of the joint distributions of SNR and cross-correlation coefficient for the three detection algorithms (EQTransformer, CRED, and FastMapSVM). Comparison of these joint distributions helps better substantiate the interpretations of the previous results with more objective analysis.

Minor Comments:

1- It would be helpful to provide more specific information about the problem the article aims to solve if the primary goal is to learn an ML model quickly and with few data sets. What kinds of problems could be solved if we used this method instead of NN? What is the importance of learning models when there are few training data sets? As an example, how much resource would be saved if we used this method for the same task done by x et al?

Thank you for this comment. We have added a sentence to the introduction that specifies example applications that would benefit from the ability to train FastMapSVM using small train data sets—viz., analyses of “icequakes,” stick-slip events at the base of landslides, and nuisance signals recorded during temporary deployments.

2- I agree with other reviewers that being intuitive here does not seem to be defined very clearly. However, I think embedding the results into similarity distance space, plotting, and visualizing them would provide a greater sense of intuition than using NNs.

I believe intuition is most effective when we can look back and see what we missed and why we missed it. You can trace back the reason for detecting an event when you use template matching, for example. The probability of false detection can be calculated using a threshold for the cross-correlation coefficient.

Here we wish to make a distinction between “intuitive” and “interpretable.” We use the term “intuitive” in the general sense of “using or based on what one feels to be true even without conscious reasoning,” whereas we explicitly define model “interpretability” as “the degree to which causal mappings between model inputs and outputs can be understood by humans.” When referring to intuition, we mean that which goes before reason. When referring to interpretability, we mean the ability to apply reason to derive understanding. What you describe above is what we refer to as “interpretability”.

The template matching process is highly interpretable because there are very few levels of transformation between the inputs and outputs, and the transformations are relatively simple. NNs, on the other hand, are difficult to interpret because they comprise many sequential, interdependent transformations. FastMapSVM is somewhere in

between, although much closer to the template matching end of the spectrum than the NN end from our perspective. Assuming one understands the meaning of the distance function employed (the correlation distance in our application to the seismogram domain, and the cosine dissimilarity in the 2-D image domain presented as Supplementary Material), the embedding process has a simple geometric interpretation based almost entirely on basic trigonometry. SVMs are somewhat more difficult to understand in a robust mathematical sense, but still have a simple geometric interpretation: A boundary is drawn in N-dimensional Euclidean space such that all points falling on one side of the boundary belong to one class and all of the points falling on the other side of the boundary belong to the other class. Precisely how SVMs derive this boundary is moderately complicated, but no more so than the means by which NNs make their decisions.

3-There are other NN models available for earthquake detection. Even though it isn't possible to compare all of them, please explain why EQTransformer and CRED were chosen. It is important to explain why only these two NN models are being compared with the method. Here is an example of a paper that claims that its objective is to have good accuracy with minimal training. O. M. Saad and Y. Chen, "Earthquake Detection and P-Wave Arrival Time Picking Using Capsule Neural Network," in IEEE Transactions on Geoscience and Remote Sensing, vol. 59, no. 7, pp. 6234-6243, July 2021, doi: 10.1109/TGRS.2020.3019520.

Thanks for suggesting this. We added a sentence describing why we choose these two models for comparison.

line 70:

Time: what does it mean? Time required for training? The structure of the NN model plays a role in this.

Yes this refers to train time. Indeed this depends on the NN architecture, but given that FastMapSVM outperforms EQTransformer, which can be considered representative of the state-of-the-art, it is highly unlikely that a NN model that is sufficiently simple to be trained faster than FastMapSVM will perform comparably.

line 108: Use math notation, 2^n where n varies from k1 to k2.

Thanks for this suggestion. It is more concise and readable. We have incorporated as suggested.

Review #3

Thanks to the authors for considering my comments to improve the paper.

I agree with the suggestions of other reviewers that this paper may be more appropriate for a seismic journal. Because there are three limitations of this work: Firstly, the algorithmic innovation of the proposed machine learning method is limited. Secondly, the main contribution of the paper lies in applying two machine learning methods (FastMap and SVM) to classify earthquakes and noise. Thirdly, the claimed advantage of better performance when training data or time is limited is not significant, because there are actually many manual labels available for earthquake detection.

Thank you for taking time to review our manuscript and provide further comments.

Indeed the novelty of the algorithm is limited in that we simply combine two existing algorithms. However, this can also be framed as a strength of the article: By combining two tried-and-true methods, we attain performance comparable to (and in certain cases better than) more complicated and novel algorithms. An important message implicit in our manuscript is that “novel” does not imply “better.”

As you suggest, the applications of the algorithm are indeed limited to “detecting earthquakes” and “identifying phases”—plus classifying images of hand-written digits in the Supplementary Material. This paper focuses on a proof of concept rather than a demonstration of versatility. A paper demonstrating the versatility of the model would form a useful follow-up; however, we want to comprehensively assess the performance of the model relative to state-of-the-art NN models. Doing so across multiple domains is beyond the scope of the current project, so we compare multiple aspects of the model against state-of-the-art models in the seismogram domain.

We respectfully disagree with the third point above. Despite the abundance of manual labels available for training models to detect earthquakes, we show in Figure 5 that FastMapSVM retains desirable performance properties despite being trained on just a fraction of the amount of data. By tailoring FastMapSVM for a particular task, it can be trained to outperform EQTransformer and CRED using a small fraction of the amount of data they require. FastMapSVM trained with 512 instances outperforms EQTransformer and CRED, despite the fact that they (EQTransformer and CRED) were trained on massive, global data sets.

I have a few minor comments on the revision:

1. Fig. 1: Why does the training time not linearly increase with the training dataset size? And why are the test times different for EQT and CREAD? From this result, it seems that the advantage of FastMapSVM is not significant. The performance and speed of other simpler models like GPD and PhaseNet can be even better.

Not that the x-axis in Fig. 1 is logarithmic (base 2). When plotted on a linear scale, we see that train time does increase linearly for CRED but not for EQTransformer (see figure below). This is likely a result of the “early stopping” rules that we apply to the training process (i.e., training ceases after either (1) a certain number of epochs, or (2) the train loss does not decrease for a certain number of epochs in a row, whichever comes first).

The main reason for the different test times for EQTransformer and CRED is that CRED applies a short-time Fourier transform to the input time series as a preprocessing step. EQTransformer does not apply this preprocessing and is thus faster.

It is true that FastMapSVM is not dramatically faster than EQTransformer or CRED, and that other simpler NN models may be faster than FastMapSVM. The key result we wish to illustrate here, however, is that FastMapSVM requires less train data and train time and has comparable execution speed to NN models.

2. Line 402: "FastMapSVM, EQTransformer, and CREAD identify, , and 13 windows with earthquakes." Numbers are missing here.

Thanks for catching this omission. These results have been replaced in response to your third comment below and a comment from another reviewer.

3. Table 1: I feel the samples are too few to make a reliable conclusion here. It should be pretty easy to apply to more earthquakes, since the Ridgecrest earthquake sequence is very active.

We agree that it is easy to apply the algorithms to more earthquakes. In the original analysis, we analyzed a small amount of data (10 minutes) because we intended to visually inspect each of the results to validate them. To better quantify the results in this section, we now measure the maximum SNR and cross-correlation coefficient (against the set of 256 template seismograms used to train FastMapSVM). By comparing the joint distributions of the maximum SNR and cross-correlation coefficient for each detection and each algorithm, we obtain a better sense of the relative performances and can analyze a larger data set. In the updated results, we analyze one full day of data.

REVIEWERS' COMMENTS:

Reviewer #2 (Remarks to the Author):

In two revised revisions, all my concerns and comments are addressed. There is just one minor comment below. Looking forward to not seeing this manuscript as a reviewer but as a reader! It is ready to go!

Line 453:

Shakibay Senobari et al (2019) provided a comprehensive explanation of cross-correlations for seismic data. It is a good idea to reference it here. It is even possible to remove a large part of this section and refer to that paper regarding the use of cross-correlation as a similarity measure for seismic data. However, that is up to the authors.

Shakibay Senobari, Nader, et al. "Super-efficient cross-correlation (SEC-C): A fast matched filtering code suitable for desktop computers." *Seismological Research Letters* 90.1 (2019): 322-334.

Reviewer #3 (Remarks to the Author):

My comments have been addressed in the revision. I think this work can be accepted for publication.

Reviewer #2

In two revised revisions, all my concerns and comments are addressed. There is just one minor comment below. Looking forward to not seeing this manuscript as a reviewer but as a reader! It is ready to go!

The authors thank the reviewer for his time in reviewing the manuscript and encouraging remarks.

Line 453:

Shakibay Senobari et al (2019) provided a comprehensive explanation of cross-correlations for seismic data. It is a good idea to reference it here. It is even possible to remove a large part of this section and refer to that paper regarding the use of cross-correlation as a similarity measure for seismic data. However, that is up to the authors.

Shakibay Senobari, Nader, et al. "Super-efficient cross-correlation (SEC-C): A fast matched filtering code suitable for desktop computers." *Seismological Research Letters* 90.1 (2019): 322-334.

The authors thank the reviewer for this suggestion. The citations in this passage of the previous version of the manuscript are application-focused; whereas the reference suggested above is method-focused. We have added the suggested reference.

Reviewer #3

My comments have been addressed in the revision. I think this work can be accepted for publication.

The authors thank the reviewer for his time.